# Prefrontal PV interneurons facilitate attention and are linked to attentional dysfunction in a mouse model of absence epilepsy

**Brielle Ferguson[1,2,3]\*, Cameron Glick[1], John R Huguenard[1]\***

[1]Department of Neurology and Neurological Sciences, Stanford University, Stanford, United States; [2]Department of Genetics, Harvard Medical School, Boston, United States; [3]Program in Neurobiology and Department of Neurology, Boston Children's Hospital, Boston, United States

**Abstract** Absence seizures are characterized by brief periods of unconsciousness accompanied by lapses in motor function that can occur hundreds of times throughout the day. Outside of these frequent moments of unconsciousness, approximately a third of people living with the disorder experience treatment-resistant attention impairments. Convergent evidence suggests prefrontal cortex (PFC) dysfunction may underlie attention impairments in affected patients. To examine this, we use a combination of slice physiology, fiber photometry, electrocorticography (ECoG), optogenetics, and behavior in the $Scn8a^{+/-}$ mouse model of absence epilepsy. Attention function was measured using a novel visual attention task where a light cue that varied in duration predicted the location of a food reward. In $Scn8a^{+/-}$ mice, we find altered parvalbumin interneuron (PVIN) output in the medial PFC (mPFC) in vitro and PVIN hypoactivity along with reductions in gamma power during cue presentation in vivo. This was associated with poorer attention performance in $Scn8a^{+/-}$ mice that could be rescued by gamma-frequency optogenetic stimulation of PVINs. This highlights cue-related PVIN activity as an important mechanism for attention and suggests PVINs may represent a therapeutic target for cognitive comorbidities in absence epilepsy.

\*For correspondence:
brielle.ferguson@childrens.harvard.edu (BF);
John.Huguenard@stanford.edu (JRH)

## Editor's evaluation

This study reports on the circuits contributing to impairment in attention in absence epilepsy linked to reduced Scn8A expression. Using a novel attentional engagement task, the evidence supporting the main conclusions is solid and well-sampled. The results provide a starting point for future experiments to assess functional impairments in parvalbumin-positive interneurons and gamma activity with attention in models of absence seizures and mental health disorders.

## Introduction

Attention is required for most aspects of moving through the world, from holding a conversation in a crowded room, to crossing a busy intersection. Disrupted attention presents as a persistent comorbidity across several disease states, even when certain hallmark symptoms respond to treatments or are in remission (*Meeren et al., 2005*; *Rock et al., 2014*). One such example is absence epilepsy, where attention deficits do not respond to current treatments and are a significant predictor of adverse long-term outcome across patients (*Masur et al., 2013*). Absence seizures are diagnosed by a distinct EEG pattern: bilateral synchronous 3–4 Hz spike-and-wave discharge (SWD), which manifests

**eLife digest** People who experience absence seizures may go through brief lapses in consciousness hundreds of times a day. They also often have difficulties engaging and remaining focused on a task, which can severely limit their ability to study, work and go through their day-to-day life. These impairments in attention persist even when medication puts a stop to the seizures, suggesting that they are not directly linked to the epileptic episodes. In fact, recent work has indicated that these deficits may be caused instead by alterations in the activity of the prefrontal cortex, the brain area which helps to regulate attention and impulsivity. However, the exact nature of these changes remains unclear, making it difficult to design treatments that could improve patients' quality of life.

To explore this question, Ferguson et al. developed a new behavioral test that allowed them to measure the attention levels of mice genetically engineered to have absence seizures. The experiments confirmed that these animals had impaired attention even when brain activity recordings showed that they were not experiencing seizures.

Further work revealed that poor performance on the behavioral test was linked to decreased activity in parvalbumin interneurons, a group of cells in the prefrontal cortex which can inhibit many other types of neurons. In mutant mice, this change was associated with alterations in network activity broadly in the cortex, including in electrical patterns which are linked to cognitive processes. Promisingly, increasing the activity of the interneurons during the attention task improved performance, suggesting that this type of cell could represent a therapeutic target for attention deficit in absence epilepsy.

as brief periods of unconsciousness accompanied by a lapse in motor function (*Gibbs et al., 1935*; *Panayiotopoulos, 1999*). Typical absence seizures are present in 10–15% of all epilepsy syndromes, and can occur hundreds of times throughout the day (*Berg et al., 1999*; *Panayiotopoulos, 2001*; *Posner, 2008*). While animal models reliably capture many aspects of the absence seizure phenotype (*Coenen and Van Luijtelaar, 1987*; *Meeren et al., 2005*; *Papale et al., 2009*; *Vergnes et al., 1982*), there has been little direct exploration of attention performance in rodent models of absence epilepsy or of the underlying mechanisms.

Components of attention include the initiation or engagement of attention, vigilance or sustained attention, and attentional shifting or flexibility (*Mirsky, 1987*). Several studies point to the dorsolateral PFC supporting many of these components in humans and primates (*Barceló et al., 2000*; *Gregoriou et al., 2014*; *Rossi et al., 2007*; *Zanto et al., 2011*). This is strengthened by lesion data from rodents of the closest functional homolog of human dorsolateral PFC, the prelimbic medial PFC (mPFC) (*Chudasama and Muir, 2001*; *Kahn et al., 2012*; *Muir et al., 1996*). Recent advances have allowed for more exploration into cell-type-specific mechanisms of attention, revealing that fast-spiking GABAergic interneurons, or parvalbumin-expressing interneurons (PVINs), show attention-related activity. Medial PFC PVINs have been shown to represent rule information in a flexible attentional shifting task (*Rikhye et al., 2018*), while population increases in PVIN activity during the delay period prior to cue onset are associated with correct choices in a sustained attention task (*Kim et al., 2016a*).

Here, we utilized in vitro local field potential (LFP) and whole-cell recordings in mPFC slices, and in vivo fiber photometry, electrocorticography (ECoG) recordings, optogenetics, and behavior to explore circuit mechanisms of attention disruption in the *Scn8a* mutant mouse model of absence epilepsy. In wild-type (WT) and *Scn8a* mutant mice, we observed that cue-evoked increases in mPFC PVIN activity occurred before correct choices during a novel attentional engagement task (AET). *Scn8a* mutant mice however, exhibited attenuated cue-evoked PVIN activity, reduced gamma band activity, and poorer performance on the AET that was independent of seizure pathology. This was associated with reduced putative feedforward inhibition in slice recordings of mPFC LFPs and increased synaptic failure rate during gamma-frequency stimulation of PVINs. Promisingly, we found that gamma frequency optogenetic activation of PVINs during the cue period was sufficient to recover attention performance in *Scn8a* mutant mice. This suggests that cue-related increases in mPFC PVIN activity support basic attentional processes and may provide both a regional and cell-type-specific therapeutic target for attention dysfunction in absence epilepsy.

## Results

### Scn8a mutant mice exhibit attention deficits on the Attentional Engagement Task

To examine the mechanisms of attention impairments in absence epilepsy, we used mice with a heterozygous loss-of-function mutation in *Scn8a (Scn8a^med)* (*Kohrman et al., 1996*) henceforward referred to as *Scn8a^+/-*. Altered expression of *Scn8a* has been observed in humans with absence seizures and cognitive impairments (*Berghuis et al., 2015*; *Trudeau et al., 2006*). *Scn8a* encodes the voltage-gated sodium channel, NaV1.6, and mice with reduced NaV1.6 expression have frequent and well-characterized absence seizures, due at least in part to specific hypofunction of inhibitory neurons in the thalamus (*Makinson et al., 2017*; *Papale et al., 2009*). Still, the implications of partial *Scn8a* loss for cortical circuit function, including in the mPFC and its role in attentional processing, remain unexplored.

To measure attention, we developed a novel assay, combining training and testing strategies from previous studies (*Kahn et al., 2012*; *Turner et al., 2015*; *Wimmer et al., 2015*). Mice (*Scn8a^+/-* and *Scn8a^+/+* littermates referred to as WT) were food-restricted to 85% of their free-feeding weight and habituated to the operant chamber. Using behavioral shaping, mice were trained to nose-poke in two distal ports to obtain a food reward (10 μL of evaporated milk). Then, mice learned to initiate trials by nose-poking in a center port, and that a visual cue, a white LED present for 5 seconds (s) in one of the two reward ports, would indicate the location of a food reward (*Figure 1A*). On average, mice learned this association in 6 training sessions, and there were no group differences in how long it took mice to reach stable performance or criterion (3 days at greater than 70% correct or 2 days greater than 90% correct, *Figure 1A*, bottom). Then, testing began, and while the cue still indicated the reward location, the cue length was varied pseudorandomly between 5 s, 2 s, 1 s, .5 s, or .1 s. We chose this range of cue lengths, hypothesizing that the long 5 s cue would require little attention due to the ease of detectability and prior exposure to this cue length during training. By contrast, the shorter cue lengths would be unexpected and significantly more difficult to report correctly, thus increasing cognitive and attentional load (*Parasuraman and Mouloua, 1987*). Additionally, by including the shortest cue lengths, .5 s and .1 s, we could approach a floor effect where most mice would have difficulty detecting the cue and we would expect them to perform poorly. With this design, we could identify specific intermediate cue lengths that would generate variable performance rates across mice and genotypes (*Dinstein et al., 2015*). We also hypothesized that performance at these intermediate cue lengths would be accompanied by related functional differences in brain activity (*Faisal et al., 2008*). To isolate basic processes related to attentional recruitment, referred to here as engagement, rather than vigilance or selective attention such as is described elsewhere (*St Peters et al., 2011*; *Young et al., 2009*), we did not incorporate any delays after trial initiation or between the cue and the response period, distractors throughout the trial, or abstraction. In all mice, we observed a performance decrement with cue length, approaching chance levels at the 0.1 s cue. However, *Scn8a^+/-* mice performed significantly worse than WT mice at intermediate cue lengths (2 s and 1 s), suggesting an impairment in attentional engagement (*Figure 1B*).

Importantly, when we kept the cue length constant within trial blocks, presumably reducing attentional load, there were no differences in accuracy between groups (*Figure 1C*). We also quantified omissions, trials in which the animal took longer than 5 s following cue termination to make a choice (likely because the animals failed to perceive the presence of the cue) and found no differences between groups (*Figure 1D*). This lack of difference in performance during single cue-length blocks or omissions suggests that observed differences on the AET are specific to increases in attentional load rather than an effect of reduced *Scn8a* on visual perception or discrimination. There was no effect of group on reaction time (seconds between cue termination and port selection); however, all mice had increased reaction time for incorrect trials, which has been correlated with impaired attention in other paradigms (*Fitzpatrick et al., 2017*; *Weissberg et al., 1990*; *Figure 1E*). To explore whether mice showed signs of behavioral inflexibility in their responses, we quantified repetitive responding, defined here as the number of blocks of five consecutive trials in which the animal selected the same port. We observed that *Scn8a^+/-* mice had a higher repetitive responding in comparison to WT littermates (*Figure 1F*). Finally, to examine whether *Scn8a^+/-* mice might exhibit broad behavioral impairments, we assayed sociability utilizing the three-chamber task and the hippocampal-dependent object recognition task and observed no significant differences between groups (*Figure 1G–L*). For subsequent AET

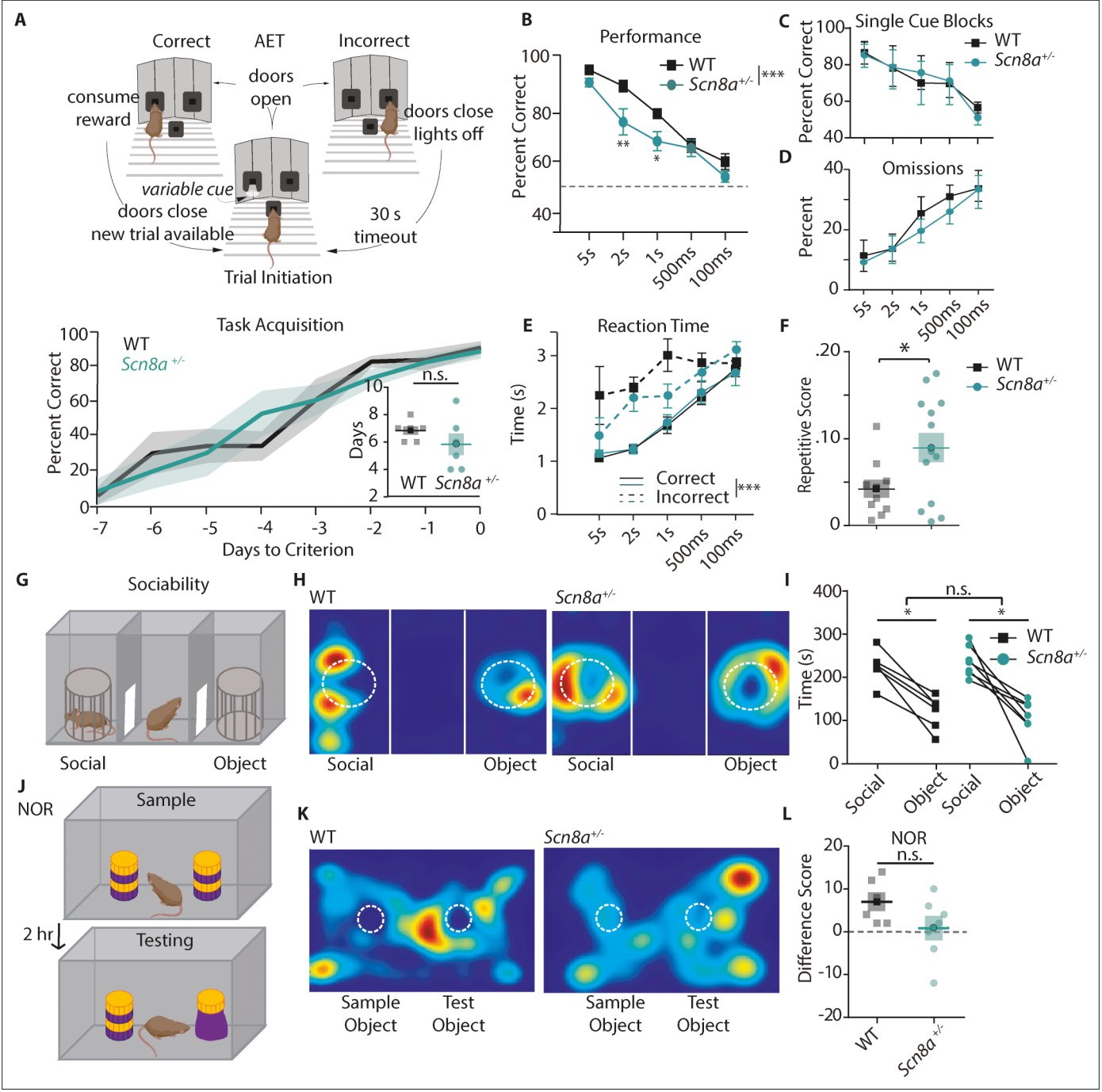

**Figure 1.** *Scn8a*[+/-] mice exhibit attention deficits on the Attentional Engagement Task (AET). (**A**) Top, AET task design. Bottom, Training accuracy for mice learning the AET. Lines represent performance displayed as average ±SE until mice reached criterion (WT are in black and *Scn8a*[+/-] are in teal for this and all subsequent figures). Inset, comparison of total days to criterion, which was not different between groups (Student's t test, p=0.27, n=7 for both groups). (**B**) *Scn8a*[+/-] mice have reduced accuracy at 2 s (p=0.0049) and 1 s (p=0.0497) compared to WT littermates (two-way ANOVA, $F_{(1,70)}$ = 16.17, p=0.0001, n=8 for both WT and *Scn8a*[+/-]). (**C**) Animals were evaluated on their performance at each cue length in separate trial blocks, and there was no effect of genotype on performance in each session ($F_{(1,40)}$=0.00077, p=0.9780, n=5 for both WT and *Scn8a*[+/-]). (**D**) Omissions were not different between groups (two-way ANOVA, $F_{(1,70)}$ = 0.8456, p=0.3609, n=8 for both WT and *Scn8a*[+/-]). (**E**) There was no effect of group on reaction time, but there was an effect of accuracy (three-way ANOVA, group, $F_{(1,70)}$ = 1.01, p=0.3161; accuracy, $F_{(1,70)}$ = 67.66, p=0.0001). (**F**) The repetitive score was increased in *Scn8a*[+/-] mice (Student's t test, p=0.0381, n=10 and 13 for WT and *Scn8a*[+/-] respectively). (**G**) Top, design of the three-chamber sociability task. (**H and K**) Representative heatmap images. (**I**) Each group showed a significant preference for the novel mouse versus the object (p<0.0001), but there were no differences between groups ($F_{(1,11)}$=0.5648, n=6 and 7 for WT and *Scn8a*[+/-] respectively). (**J**) Novel object recognition paradigm. (**L**) There were no significant differences in the difference score (Student's t test, p=0.111, n=6 and 7 for WT and *Scn8a*[+/-] respectively). Data are shown as averages from individual animals or groups, and errors bars or shading represent ±SE in this and all future figures unless stated otherwise.

experiments we utilized only three cue lengths in our analysis, 5 s as the long cue, 2 s as our inter-mediate (int) attention-related cue, and 0.5 s as the short cue to simplify the task interpretation, data presentation, and reduce the number of statistical comparisons.

## AET performance deficits are unrelated to acute seizure activity in Scn8a$^{+/-}$ mice

Absence epilepsy patients experience attention impairments during interictal periods and even when seizures are well controlled with anti-epileptic drugs (*Barone et al., 2020*), suggesting that atten-tion dysfunction is not purely due to the periodic losses of consciousness associated with SWDs. Additionally, absence seizure prevalence is highest in periods of quiet waking (*Barone et al., 2020*). Thus, we hypothesized that seizure burden would be low while animals were actively performing the task and would not influence task performance. To examine this, we implanted Scn8a$^{+/-}$ mice with screws for cortical surface (ECoG) recordings over the seizure-prone somatosensory cortex (*Fogerson and Huguenard, 2016*) and recorded ECoG signals while animals were either engaged in the task (*Figure 2A and B*) or in their home cage. Engaging in the task reduced seizure frequency in Scn8a$^{+/-}$ mice in comparison to that recorded in their home cage (*Figure 2C*). To assess whether SWDs may contribute to reduced performance in Scn8a$^{+/-}$ mice, we compared the ECoG spectral power using wavelet decomposition (*Torrence and Compo, 1998*), between correct trials and incorrect trials (*Figure 2D and E*). We found no difference in the Ø (7–10 Hz) or β (10–20 Hz) range (*Figure 2F and G*) which encompass the fundamental and 2$^{nd}$ harmonic frequency bands for absence seizures, respectively (*Sorokin et al., 2017*; *Figure 2H*). In Scn8a$^{+/-}$ mice, there were no differences in power across cue lengths (*Figure 2I–K*). As we would expect a spectral power *increase* with any seizure activity (example SWD, *Figure 2H*), the lack of difference in seizure band power during correct versus incorrect trials suggests that seizures do not contribute to AET errors in Scn8a$^{+/-}$ mice. On average, in Scn8a$^{+/-}$ mice, ~1% of trials had seizures within 10 s of the trial, and in fewer than 1% of trials (0.24%) were we able to detect seizures during the cue (*Table 1*). Finally, power in seizure-related frequency bands was not different between Scn8a$^{+/-}$ mice and WTs for the intermediate cue (*Figure 2L*), where the relative difference in performance was greatest. This indicates that SWDs are largely absent while animals perform the AET, and differences in absence seizure activity do not explain reductions in performance in Scn8a$^{+/-}$ mice.

## Photoinhibition of mPFC during cue presentation reduces accuracy in the AET but not the VDST

Given that our task is a novel adaptation of previous cue-based attention tasks, we wanted to explore whether the AET relied upon the mPFC, an area classically linked to attention across multiple species (*Chudasama and Muir, 2001*; *Everling et al., 2002*; *Kastner and Ungerleider, 2000*). To do so, we implanted bilateral optical fibers targeting the prelimbic mPFC in mice with endogenous expression of Channelrhodopsin-2 (ChR2) in neurons expressing the vesicular GABA transporter (VGAT:ChR2). As optogenetic activation of interneurons has been shown to inactivate cortical network activity (*Guo et al., 2014*), this would allow for mPFC inhibition during distinct task epochs (*Figure 3*). First, we confirmed in vitro that blue light could inhibit neuronal activity continuously throughout the durations used in the AET (*Figure 3—figure supplement 1*). We collected mPFC slices from VGAT:ChR2 mice, in which blue light will activate GABAergic interneurons. We injected a noisy current barrage and recorded evoked spikes from Layer 2/3 pyramidal neurons in baseline (no light) and light conditions using a 5 s, 2 s, and 0.5 s continuous pulse (*Figure 3—figure supplement 1*). Then, we measured spikes in the last 0.5 s bin for each condition and found a significant reduction at each pulse length (*Figure 3—figure supplement 1*). Similarly, we observed that continuous blue light can hyperpolarize neurons at the same pulse lengths (*Figure 3—figure supplement 2*).

Once we validated that blue light delivery to VGAT:ChR2 mice could modulate mPFC pyramidal neuron activity, we trained mice on both the AET (*Figure 3A*) and a modified variable delay-to-signal task (VDST, *Leite-Almeida et al., 2013*) for comparison (*Figure 3B*). In the AET, we silenced mPFC during the variable cue period (*Figure 3C*) where we hypothesized attention is most heavily engaged, and observed a small reduction in performance with continuous light stimulation (*Figure 3D*) without a significant change in reaction time. To assess whether our reduction in performance could be reflecting an inability to report the cue in a matter unrelated to attentional load, we utilized the VDST,

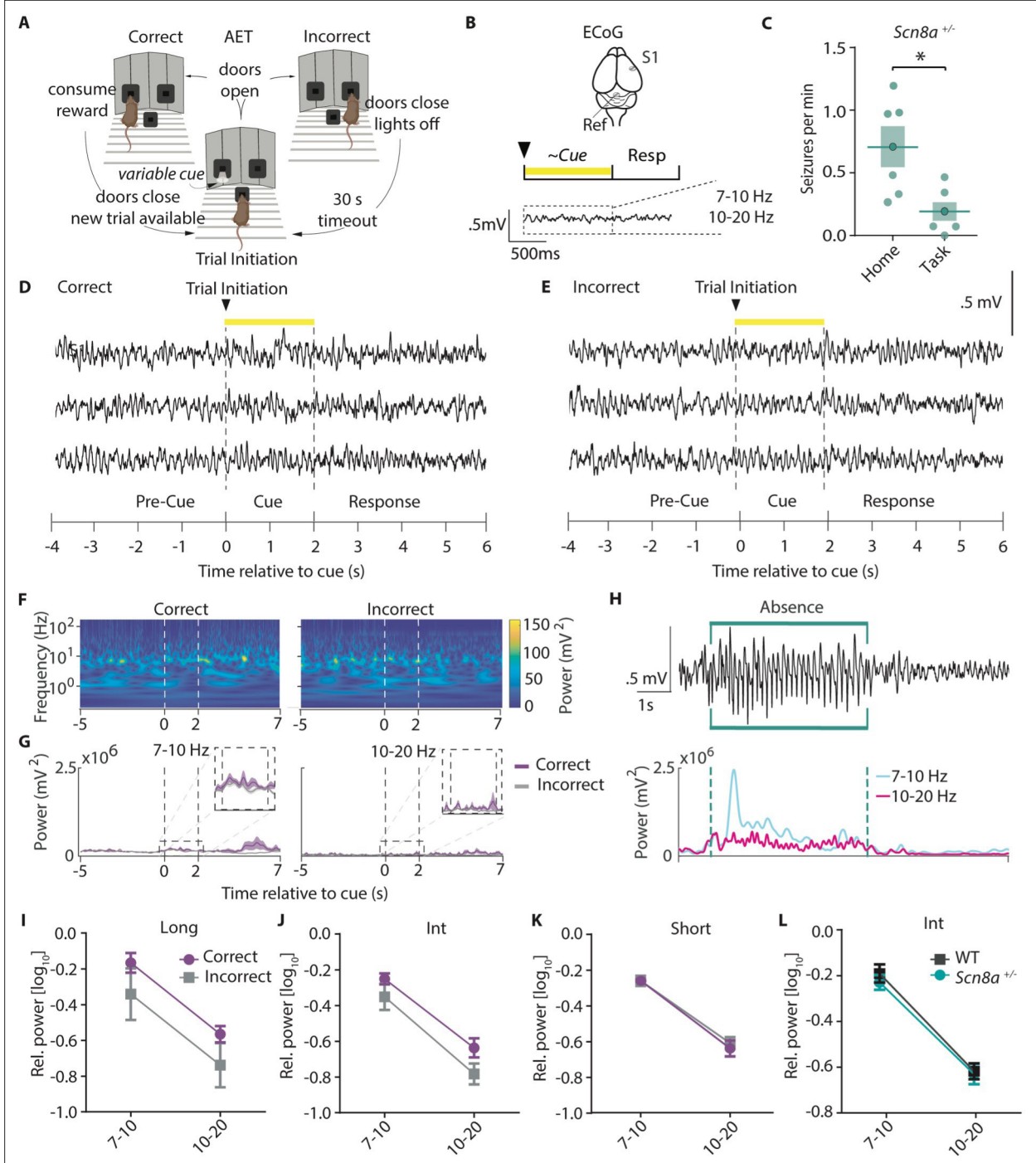

**Figure 2.** AET performance deficits are unrelated to acute seizure activity in *Scn8a*[+/-]mice. (**A**) AET design. (**B**) Illustration of ECoG recording and metrics for power analysis. (**C**) Seizures were reduced during the task (Student's T test, p=0.0155, n=6 and 6 for home cage versus task animals respectively). (**D-E**) Example ECoG activity from the highly seizure-active somatosensory cortex (S1) recorded during correct and incorrect trials with a 2 s cue length. (**F**) Representative time-frequency wavelet decomposition for correct trials (left) and incorrect trials (right). (**G**) Trial-averaged power and variance over time in the Ø (7–10 Hz) or β (10–20 Hz) range with correct trials in purple and incorrect trials in grey. (**H**) An example of a detected absence seizure (top) and the corresponding power in the Ø (7–10 Hz) or β (10–20 Hz) range. (**I-K**) In *Scn8a*[+/-] mice, there were no differences in power across cue lengths in the Ø (7–10 Hz) or β (10–20 Hz) range between correct and incorrect trials (I, Long cue, two-way ANOVA, accuracy effect, $F_{(1, 16)}$=3.998, p=0.0628; (**J**) Int cue, two-way RM ANOVA, accuracy effect, $F_{(1, 10)}$=3.367, p=0.0964; (**K**) short cue, two-way RM ANOVA, accuracy effect, $F_{(1, 10)}$=0.2082, p=0.6580; n=6 for all) (**L**) There was no difference in average power between WT and *Scn8a*[+/-] mice in the Ø (7–10 Hz) or β (10–20 Hz) range (two-way ANOVA, group effect, $F_{(1, 18)}$=0.1185, p=0.5101, n=5 and 6 for WT and *Scn8a*[+/-] mice, respectively). See also .

**Table 1.** Quantification of seizure frequency during the AET.
Seizures/min (example seizure in *Figure 2*) during the task, percent of trials with seizures during cue, and percent of trials with seizure within 10 s of trial are presented. All measurements represent average totals for animals across three behavioral sessions. N=6 for all measurements.

| Measure | Average | SE |
| --- | --- | --- |
| Seizures/min | 0.19 | 0.07 |
| Trials with seizure within 10 seconds of cue(%) | 0.44 | 0.29 |
| Trials with seizure during cue (%) | 1.94 | 1.06 |

where we would predict attentional load should be greater in a different task epoch, the delay. When we photo-inhibited mPFC during the fixed cue period in the VDST (*Figure 3E*), there were no differences in the percentage of correct choices or reaction time (*Figure 3F*). However, when we silenced mPFC activity during the variable delay period (*Figure 3G*), light stimulation reduced performance (*Figure 3H*), without a significant difference in reaction time. This suggests that the AET is linked to the mPFC, and that mPFC activation plays a role during cue presentation in an attention-dependent manner.

## Scn8a$^{+/-}$ mice exhibit a loss of post-synaptic signaling in mPFC slice recordings

Our optogenetic experiments indicated that the AET is linked to mPFC function, suggesting a potential frontal locus for performance impairments in *Scn8a$^{+/-}$*mice. In patients, SWD is prominent over the frontal cortex, also suggesting frontal dysfunction (*Archer et al., 2003*; *Tucker et al., 2007*). Additionally, individuals with absence epilepsy exhibit reduced functional activation of the PFC that correlates with attentional impairments (*Killory et al., 2011*). Given this, we began with an in vitro approach to test for *Scn8a$^{+/-}$*related dysfunction in the prelimbic mPFC. We recorded network activity using evoked local field potentials (LFPs) in coronal slices collected from *Scn8a$^{+/-}$*mice and WT littermates (*Figure 4A and B*) and derived a measure of current source density (CSD, *Figure 4C*; *Freeman and Nicholson, 1975*). This revealed several sinks or regions where cations flow into cells and out of the extracellular space, such as at excitatory synapses. We also observed nearby current sources, commonly attributed to the return flux of positive ions out of cells and back to the extracellular space. We used a pharmacological approach to identify these distinct CSD components (*Figure 4D–H*, *Figure 4—figure supplement 1*), and found a significant reduction specifically in one feature – a late source present over Layer 2/3 (*Figure 4H*). Pharmacological characterization indicated that this source represents a disynaptic response given its sensitivity to post-synaptic blockers, DNQX and CPP (*Figure 4—figure supplement 1*). Of note, sources can indicate sites of inhibitory synapses where anions flow into cells, with nearby sinks reflecting return paths. Thus, we hypothesized that *Scn8a$^{+/-}$*mice may have attenuated recruitment of local mPFC GABAergic interneurons.

## PVINs show deficient gamma-frequency synaptic output but can regulate pyramidal neuron spiking

PVINs have been classically linked to feedforward inhibition, a type of disynaptic inhibition important for gating cortical responses (*Delevich et al., 2015*; *Swadlow, 2003*), as well as successful attentional performance and cognitively associated gamma oscillations (*Cardin et al., 2009*; *Kim et al., 2016a*; *Sohal et al., 2009*). Thus, we chose to examine their regulation of mPFC pyramidal neuronal activity using whole-cell recordings (*Figure 5A*). We utilized *Scn8a$^{+/-}$*and WT mice expressing ChR2 endogenously in PVINs (PV:Ai32; *Scn8a$^{+/-}$*or PV:Ai32;WT). Given the link between PVINs and gamma frequency activity (*Cardin et al., 2009*; *Sohal et al., 2009*)**,** we stimulated PVINs with brief 0.5 s 40 Hz blue light trains and recorded the evoked inhibitory post-synaptic potential (IPSP) train in Layer 2/3 pyramidal neurons at rest in current clamp mode (*Figure 5B*). We measured synaptic failures by looking at the percentage of IPSPs within the train that were smaller than a 0.25 mV threshold to determine a failure rate for each neuron. We recorded significantly more failures in *Scn8a$^{+/-}$*mice in comparison to WT (*Figure 5C*). Additionally, to measure whether gamma-frequency activation of PVINs was capable of regulating pyramidal neuron spiking, we gave a 0.5 s noisy current injection as described above to

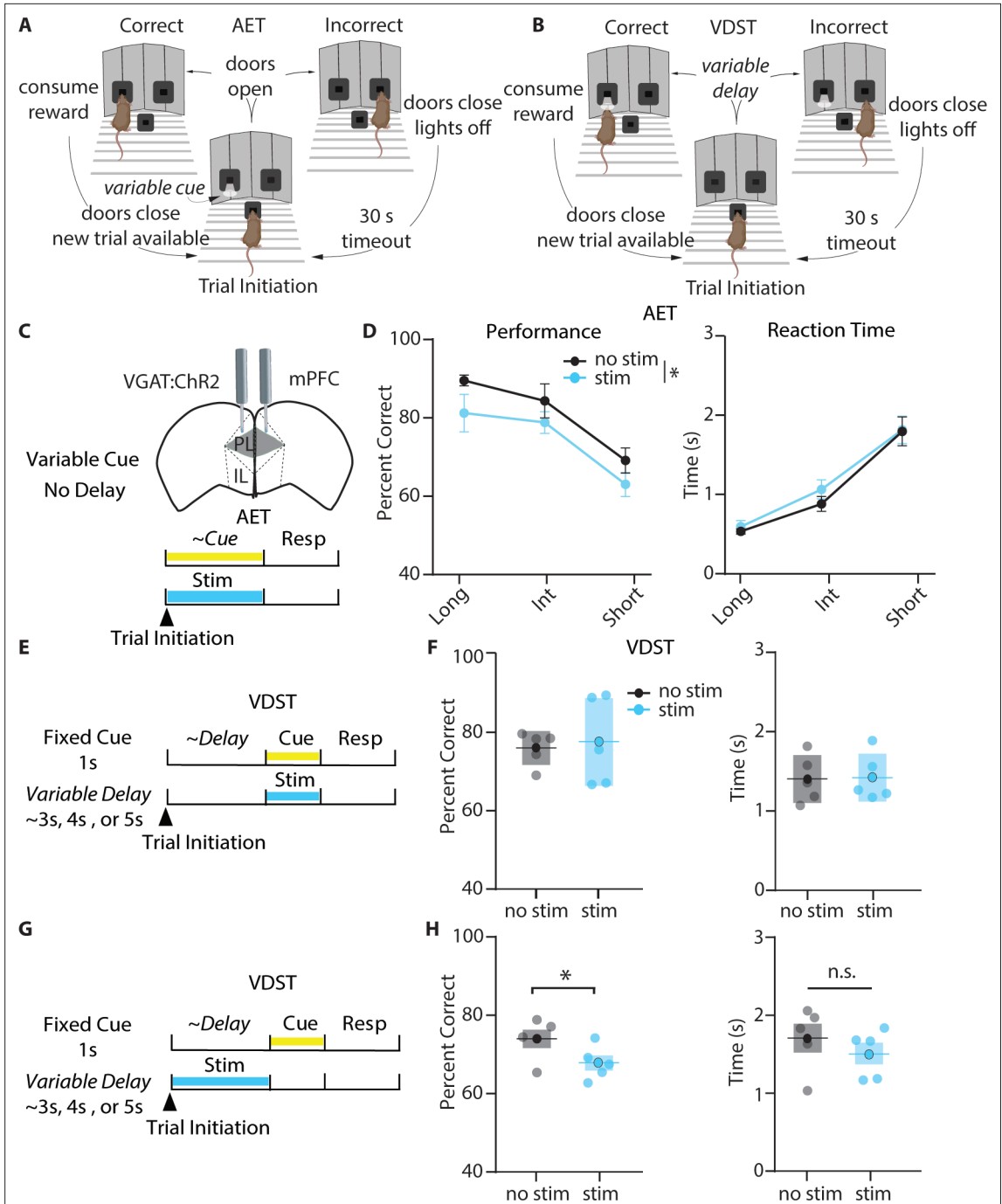

**Figure 3.** Photoinhibition of mPFC during cue presentation reduces accuracy in the AET but not the VDST. (**A-B**) Task design for the AET and VDST task. (**C**) Experimental approach for disrupting prelimbic mPFC activity during the variable cue period with continuous blue light in VGAT:ChR2 mice. (**D**) Photoinhibition with blue light reduces performance (left, two-way RM ANOVA, group effect $F_{(1,18)} = 6.990$, p=0.0165, n=6), without affecting reaction time (right, two-way RM ANOVA, $F_{(1,18)} = 3.323$, p=0.0850, n=6) in the AET. (**E**) Approach for photoinhibition during the fixed cue period in the VDST task. (**F**), Neither performance nor reaction time is affected by light stimulation during the cue in the VDST (left, performance, paired t test, p=0.7695; right, reaction time, paired t test, p=0.6508; n=6 for both). (**G**) Approach for delivering continuous light stimulation during the variable delay period. (**H**) Stimulation during the delay reduces performance in the VDST (left, paired t test, p=0.0140) without altering reaction time (right, paired t test, p=0.1268). See also *Figure 3—figure supplements 1 and 2*.

The online version of this article includes the following figure supplement(s) for figure 3:

**Figure supplement 1.** Continuous blue light can inhibit pyramidal neuronal spiking at pulse durations relevant to the AET.

**Figure supplement 2.** Continuous blue light can hyperpolarize pyramidal neurons at pulse durations relevant to the AET.

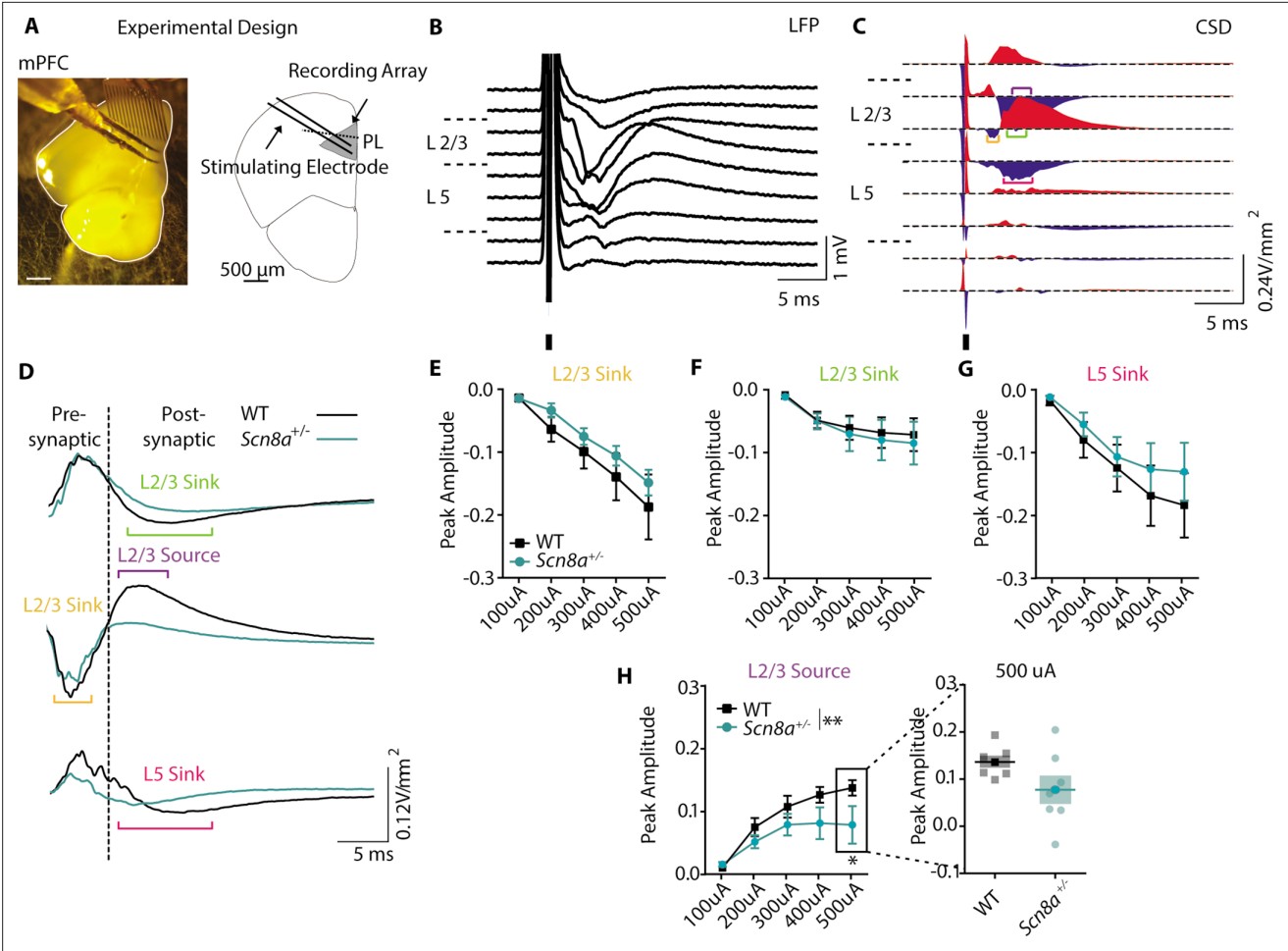

**Figure 4.** $Scn8a^{+/-}$ mice exhibit a loss of post-synaptic signaling in mPFC slice recordings. (**A**) Experimental design of LFP recordings. (**B**) Representative local field potential (LFP) recording in response to a 250 µA electrical pulse. (**C**) Representative current source density (CSD) derived from the LFP recording. Brackets highlight the traces used for quantification (orange = L2/3 sink, purple = L2/3 source, green = L2/3 sink, magenta = L5 sink). (**D**) Representative responses of the pre- and post-synaptic L2/3 sink, L2/3 source, and L5 Sink to 250 µA electrical stim. (**E-H**) Peak amplitude curves (V/mm²) of the response to electrical stim with increasing intensity. (**H**) Only the L2/3 source showed a significant reduction in peak amplitude (two-way ANOVA, group effect, $F_{(1, 60)}$=8.125, p=0.0060, n=7 slices for both WT and $Scn8a^{+/-}$ mice). See also *Figure 4—figure supplement 1*.

The online version of this article includes the following figure supplement(s) for figure 4:

**Figure supplement 1.** DNQX and CPP reduce late activity in mPFC LFP recordings.

evoke spikes and measured the spike number without light in comparison to that recorded with a 40 Hz stimulus train (*Figure 5D*). We observed that the spike reduction rate with 40 Hz PVIN activation was retained in $Scn8a^{+/-}$ mice (*Figure 5E*), indicating that PVINs are still capable of inhibiting pyramidal neuron output when stimulated optogenetically.

## Cue-evoked PVIN activity is reduced in Scn8a⁺/⁻ mice during the AET

To determine the participation of PVINs during behavior, we utilized fiber photometry to capture bulk $Ca^{2+}$ signals. We injected an AAV5-CAG-FLEX-GCaMP6s virus into the mPFC of $Scn8a^{+/-}$ mice and WT littermates with Cre expression in PVINs (PV-Cre;$Scn8a^{+/-}$ and PV-Cre;WT) and implanted an optical fiber for light delivery and collection (*Figure 6A*). Following recovery from surgery, animals were trained on the AET as described above and GCaMP signals were collected throughout testing. The continuous signal (*Figure 6B*) was segmented and aligned to trial initiation, and trials were sorted by cue length and accuracy (representative individual trials and average activity, *Figure 6C and D*). First, we took an average of each animal's change in fluorescence (dF/F) during the baseline (the second immediately prior to the cue) and throughout the cue period for correct and incorrect trials and then

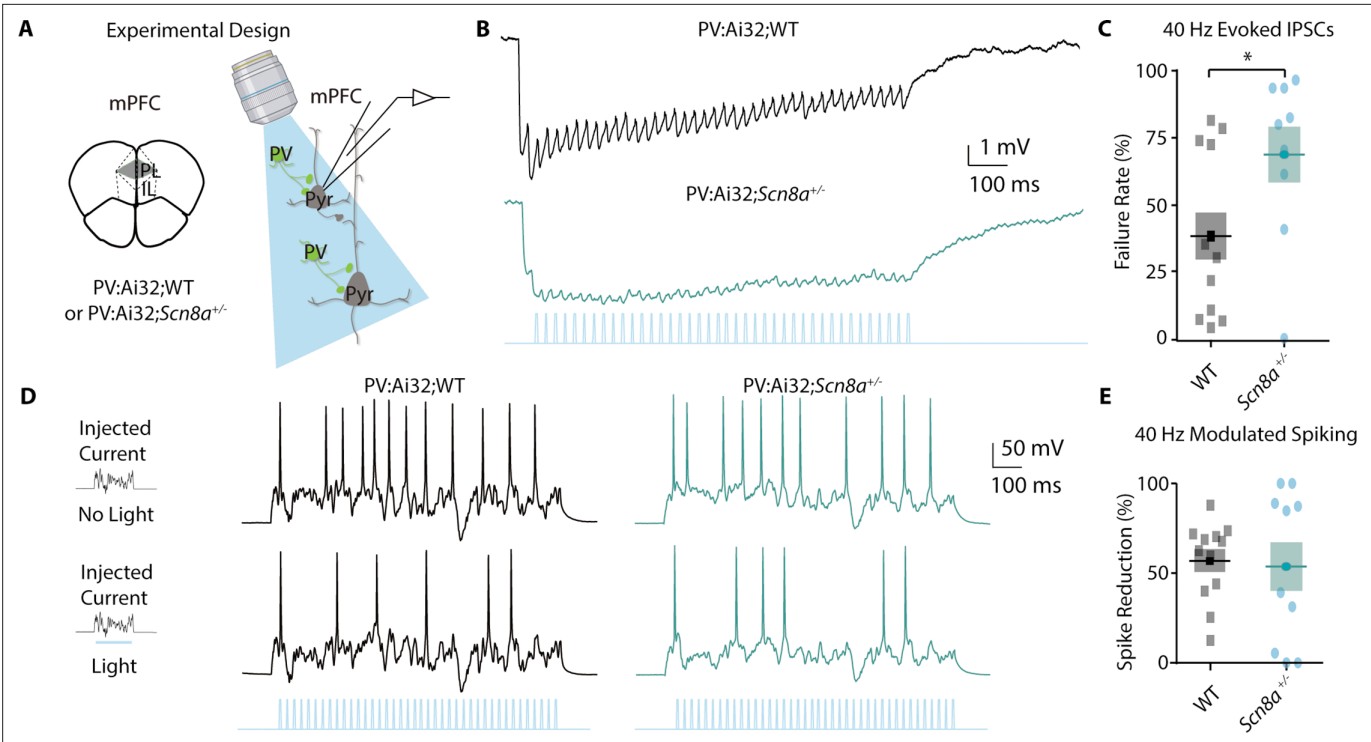

**Figure 5.** PVINs show deficient gamma-frequency synaptic output but can regulate pyramidal neuron spiking. (**A**) Experimental approach for using blue light to activate PV-positive neurons and record from pyramidal neurons. (**B**) Representative evoked inhibitory post-synaptic potential train (IPSP) to a 40 Hz train of blue light pulses. (**C**) The percentage of synaptic failures (responses less than 0.25 mV) within the 40 Hz train was significantly increased in *Scn8a*+/- mice (Mann-Whitney test, p=0.0396; n=12 and 9 for WT and *Scn8a*+/-, respectively). (**D**) Representative evoked spikes from a noisy current injection in no light (top) and 40 Hz stimulation (light, bottom) conditions. (**E**) WT and *Scn8a*+/- mice showed no differences in the percent spike reduction in response to 40 Hz stimulation (unpaired t test, p=0.8123; n=12 and 10 for WT and *Scn8a*+/- respectively).

compared cue-related activity for all cue lengths. We found that for correct trials PVIN activity was consistently modulated only at intermediate cue lengths for both groups (*Figure 6E*). However, for incorrect trials, we found no significant modulation in PVIN GCaMP signals at any cue length.

Increases in PVIN activity seemed to be related to accuracy as it was observed during correct trials at the intermediate cue lengths for both groups. When we compared cue activity between WT and *Scn8a*+/-mice, only for the intermediate cue was average dF/F decreased in *Scn8a*+/-mice (*Figure 6F*). The peak amplitude was also significantly reduced in *Scn8a*+/-mice at long and intermediate cues, but not the short cue (*Figure 6—figure supplement 1A*). This suggests that higher average and peak PVIN activity may be related to accuracy, as these measures increase in both genotypes during the intermediate cue where we hypothesize attention is most heavily and variably recruited. It is also reduced on average in *Scn8a*+/-mice specifically at the cue length where we observed performance impairments. Time to peak PVIN activity was not different between groups (*Figure 6—figure supplement 1B*; *Table 1*).

Next, we asked whether measures of cue-related PVIN activity could predict accuracy on a trial-by-trial basis. We trained group-specific classifiers using randomly selected training data with features (*Figure 6—source data 1*) from all trials within each group and across cue lengths and evaluated its performance on a subset of held-out data. Across (*Figure 6G*) and within (*Figure 6—figure supplement 2*) cue lengths for both groups the models could identify accurate trials, as determined by receiver operating characteristic area under the curve scores (ROC-AUC, *Fawcett, 2006*). This suggests that broadly, features of cue-related PVIN activity patterns are linked to the accuracy of the upcoming choice.

## Optogenetic activation of PVINs improves performance in Scn8a+/- mice while common AED, VPA has no effect

PVINs are critical in the generation of gamma oscillations (*Cardin et al., 2009*; *Sohal et al., 2009*) and activation of PVINs in the gamma range has produced beneficial effects on attention and cognition

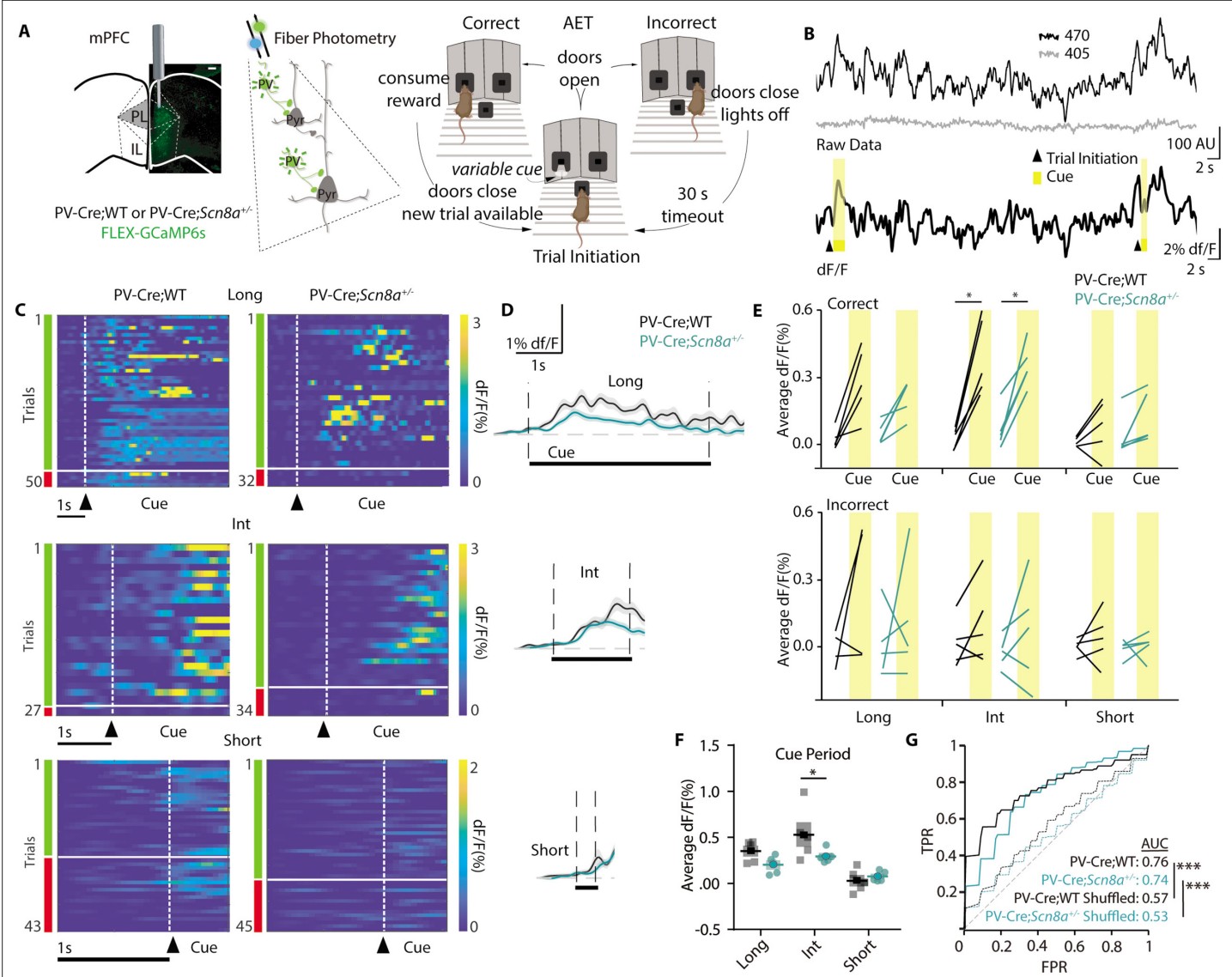

**Figure 6.** Cue-evoked PVIN activity is reduced in *Scn8a*⁺/⁻ mice during the AET. (**A**) PV-Cre; WT and PV-Cre;*Scn8a*⁺/⁻ were injected with a FLEX-GCaMP6s virus in the mPFC (scale bar = 500 µm), and tested on the AET while recording Ca²⁺ signals from PVINs using fiber photometry. (**B**) Top, example of the raw data collected from 470 (Ca²⁺-dependent) and 405 (Ca²⁺-independent isosbestic) excitation. Bottom, the same data trace after calculating the dF/F, with trial initiation (triangle) and cue presentation (yellow bar) indicated. (**C**) Heat maps of all trials of dF/F activity and (**D**) average dF/F activity from representative animals in both groups and across cue lengths. (**E**) Average dF/F baseline and cue activity at each cue length with groups indicated by line color. There was an effect of accuracy, (two-way RM ANOVA F $_{(1, 45)}$=10.372, n=5 per group) and cue (two-way RM ANOVA F $_{(1, 45)}$=41.967, n=5 per group) on PVIN activity. PVIN activity increased during the int cue in correct trials (p=0.0138 and 0.0276 for PV-Cre;*WT* and PV-Cre;*Scn8a*⁺/⁻ respectively). (**F**) Average PVIN activity was reduced during the intermediate cue (p=0.0466) for *PV:Cre;Scn8a*⁺/⁻ mice (two-way ANOVA, F $_{(1,24)}$=4.658, p=0.0411; n=5 per group). (**G**) Classifiers trained on features from the FiP cue data across cue lengths could predict trial outcome within groups (AUC test set performance vs shuffled data; p=0.00022 and 0.00020 for PV-Cre;*WT* and PV-Cre;*Scn8a*⁺/⁻, respectively). See also *Figure 6—figure supplements 1 and 2*.

The online version of this article includes the following source data and figure supplement(s) for figure 6:

**Source data 1.** Features used to train the classifiers for predicting trial outcome from PVIN activity.

**Figure supplement 1.** Peak Amplitude is reduced at the long and intermediate cue lengths, while time to peak dF/F is not.

**Figure supplement 2.** Trial outcome can be predicted by PVIN activity for individual cue lengths.

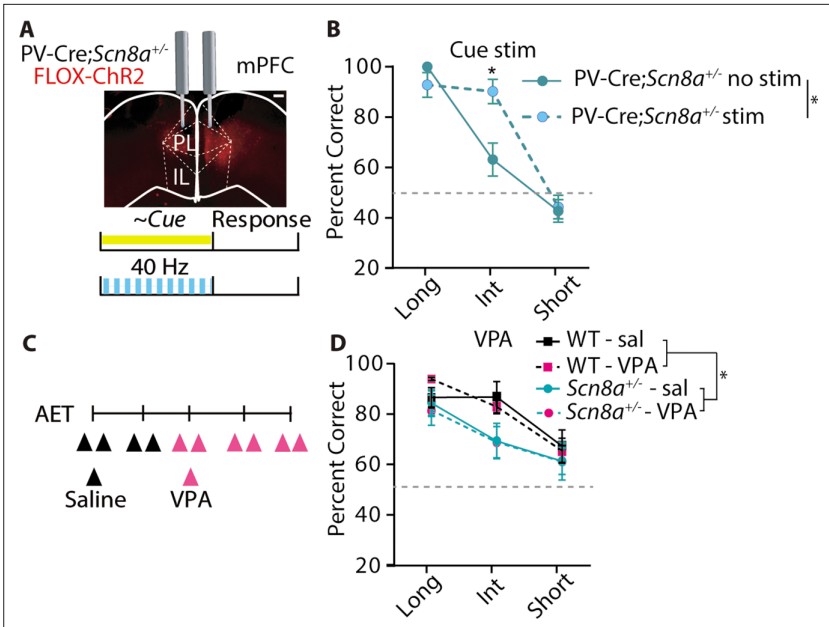

**Figure 7.** Optogenetic activation of PVINs improves performance in *Scn8a*[+/-]mice while common AED, VPA has no effect. (**A**) Mice received a bilateral injection with the FLOX-ChR2 virus (scale bar = 500 μm), and then were implanted bilaterally with optical fibers for delivering blue light during the cue (40 Hz, 5ms pulse width). (**B**) With gamma stimulation, there is a significant effect of light (two-way ANOVA, $F_{(1, 22)}$=6.973, p=0.0173, n=7 mice) during the intermediate cue (p=0.0044). (**C**) Timeline of saline and VPA injections (200 mg/kg) during AET testing. (**D**) While *Scn8a*[+/-] *mice* had reduced performance in comparison to WT, there was no effect of VPA on performance (three-way ANOVA, group effect, $F_{(1, 35)}$=5.850, p=0.0191, VPA effect, $F_{(1, 35)}$=0.010, p=0.943; n=3 and 6 for WT and *Scn8a*[+/-] mice, respectively). See also *Figure 7—figure supplements 1 and 2*.

The online version of this article includes the following figure supplement(s) for figure 7:

**Figure supplement 1.** 40 Hz stimulation increases mPFC ECoG power in the 30–50 Hz range.

**Figure supplement 2.** 40 Hz stim has no effect on animals expressing the control virus and seizures are not significantly reduced by VPA.

(*Cho et al., 2015*; *Cho et al., 2020*; *Kim et al., 2016a*). Given the observed reduction in performance at the intermediate cue in *Scn8a*[+/-]mice, we hypothesized that supplementing gamma optogenetically may improve attention. Thus, we expressed channelrhodopsin2 (AAV5-FLOX-ChR2-mCherry) in mPFC PVINs of PV-Cre;*Scn8a*[+/-] mice and implanted optical fibers for light delivery (*Figure 7A*). Activation of PVINs at gamma frequency (40 Hz) increased the percentage of correct choices for PV-Cre;*Scn8a*[+/-] mice specifically at the intermediate cue (*Figure 7B*). We also observed that 40 Hz stimulation was capable of increasing ECoG power in the 30–50 Hz range in a separate cohort of mice *Scn8a*[+/-] mice expressing ChR2 in PVINs (*Figure 7—figure supplement 1*). Importantly, this effect was not observed in PV-Cre;*Scn8a*[+/-] mice that underwent the same experimental protocol with a control eYFP virus (*Figure 7—figure supplement 2A–B*).

We were also curious whether reducing seizures would improve performance in the AET, given the lack of efficacy in mitigating cognitive symptoms in patients even when their seizures are controlled (*Figure 7C*; *Conant et al., 2010*; *D'Agati et al., 2012*). WT and *Scn8a*[+/-]mice were treated twice daily with saline or the anti-epileptic drug, Valproic Acid (VPA), the latter of which significantly reduced seizure burden in *Scn8a*[+/-]mice (*Figure 7—figure supplement 2D*). Although *Scn8a*[+/-]mice exhibited reduced performance compared to WTs, there was no effect of VPA on accuracy in either group (*Figure 7D*).

## *Scn8a*[+/-] mice exhibit reductions in mPFC gamma power along with AET performance deficits

Our previous findings indicated reductions in cue-evoked PVIN activity and that 40 Hz optogenetic stimulation of PVINs could improve attention dysfunction in *Scn8a*[+/-] mice. Given that analysis of ECoG/

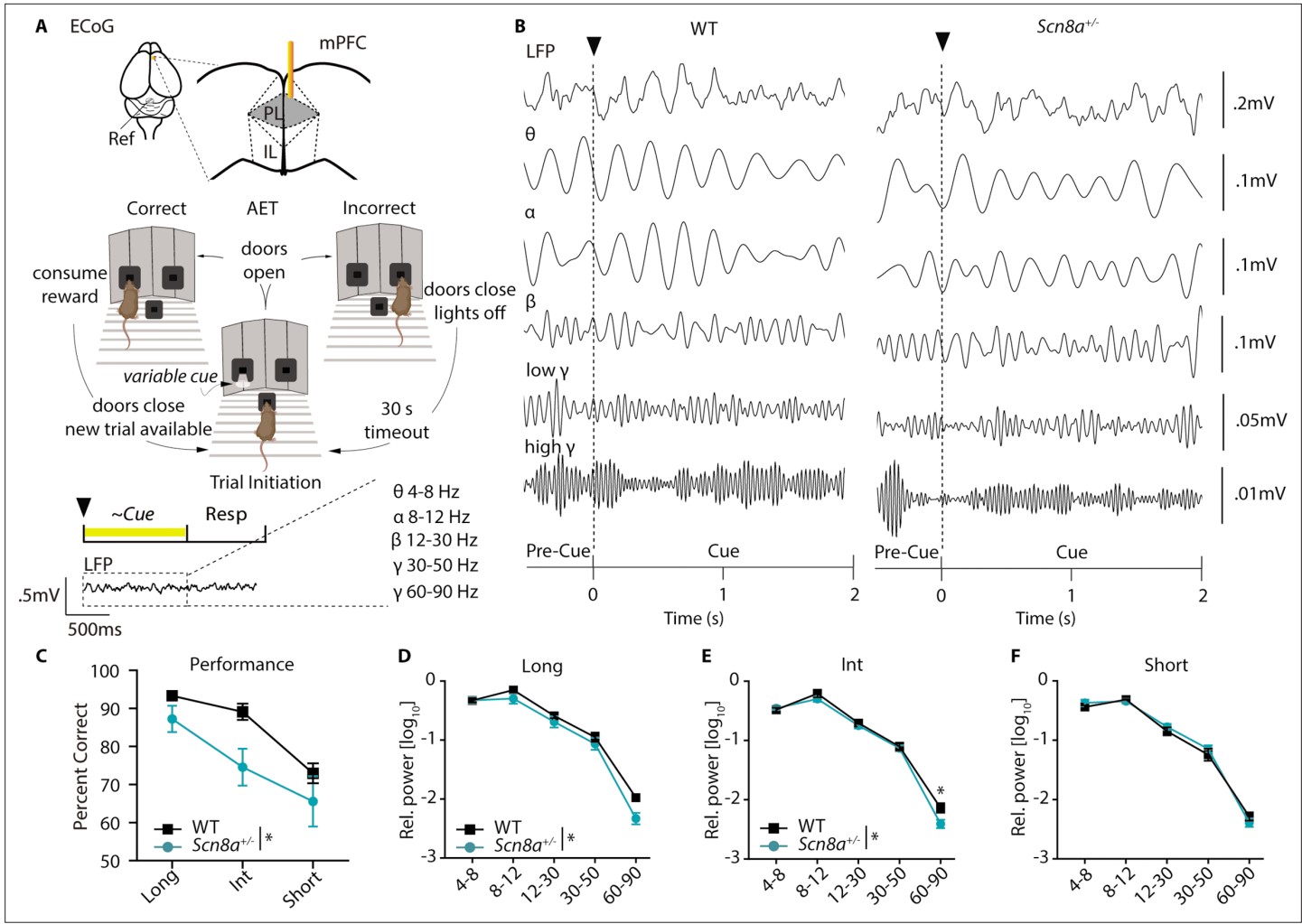

**Figure 8.** *Scn8a*[+/-] mice exhibit reductions in mPFC gamma power along with performance deficits in the AET. (**A**) Experimental design, ECoG activity was recorded using a gold electrode over mPFC and a reference screw over the cerebellum during the AET. Power was measured during the cue period across frequency bands. (**B**) Representative ECoG filtered across different frequency bands during the cue period. (**C**) Performance was decreased in *Scn8a*[+/-] mice (two-way ANOVA, $F_{(1, 21)}=4.497$, p=0.0460, n=3 and 6 for WT and *Scn8a*[+/-] mice, respectively). (**D-F**) *Scn8a*[+/-] mice have reduced power during the long (high gamma, p=0.0597) and int cues (high gamma, p=0.0145), but not during the short cue (long, two-way ANOVA, $F_{(1, 35)}=5.623$, p=0.0234; int, two-way ANOVA, $F_{(1, 35)}=4.586$, p=0.0393; short, two-way ANOVA, $F_{(1, 35)}=0.2656$, p=0.6095; n=3 and 6 for WT and *Scn8a*[+/-] mice, respectively).

EEG remains the gold standard for diagnosis of absence epilepsy, we wondered if ECoG recordings might facilitate identification of network-level biomarkers related to attention dysfunction that could potentially be measured in humans. We implanted a cohort of *Scn8a*[+/-] and WT mice with a gold electrode over the mPFC and recorded ECoG (*Figure 8A*) while mice performed the AET. As described above, we performed wavelet decomposition to obtain ECoG spectral power across different frequency bands (*Figure 8B*) and compared power between *Scn8a*[+/-] mice and WT littermates. AET performance was reduced in *Scn8a*[+/-] mice (*Figure 8C*) with concomitant reductions in spectral power for both the long and intermediate cue (*Figure 8D and E*). Specifically for the intermediate cue, there was a significant reduction in high gamma activity (60–90 Hz) in *Scn8a*[+/-] mice. For the short cue we observed no differences in power across any frequency band (*Figure 8F*), which is consistent with the absence of behavioral differences or evoked PVIN activity at this cue length.

## Discussion

Our findings demonstrate that increases in cue-related mPFC PVIN activity are associated with correct choices during a novel attentional engagement task (AET). Additionally, PVIN activity is reduced in *Scn8a*[+/-]mice along with poorer performance on the AET. These results also provide the first evidence of both the behavioral manifestation and a circuit biomarker of attention impairments in an animal model of absence epilepsy, one of the most commonly presenting cognitive comorbidities associated with the syndrome. Further, our results suggest that the recruitment of PVINs and gamma activity represent key circuit mechanisms during attentional engagement.

### Attentional and cognitive impairments in absence epilepsy

Absence epilepsy is classically defined as an interruption of consciousness due to the emergence of stereotyped spike-and-wave activity ranging from 3 to 6 Hz in clinical populations (*Watanabe, 2003*). However, patients also present with profound cognitive and behavioral impairments (*Caplan et al., 2008*; *Pavone et al., 2001*; *Williams et al., 1996*) with attention being one of the most severely impacted comorbidities (*Fonseca Wald et al., 2019*). These attention impairments are present even in interictal periods (*Conant et al., 2010*; *D'Agati et al., 2012*; *Vega et al., 2010*) and do not respond to current anti-epileptic drugs (*Glauser et al., 2010*; *Masur et al., 2013*). Mutations in *Scn8a* have been observed in patients with absence epilepsy, intellectual disability, attention-deficit hyperactivity disorder, and autism spectrum disorders (*Liu et al., 2019*; *Papale et al., 2009*; *Trudeau et al., 2006*; *Wagnon et al., 2017*). Across these syndromes, attention impairments represent a convergence point, and gaining a better understanding of attentional dysfunction is critical for improving quality of life in people living with these disorders. Thus, this study focused on investigating underlying mechanisms in an *Scn8a*[+/-]mouse model that captures aspects of human generalized absence epilepsy (*Berghuis et al., 2015*).

Using the AET, we confirmed that the *Scn8a*[+/-]model bore several similarities to the human condition. Most importantly, we observed a consistent impairment in attention that appeared to exist independent of any acute effects of the seizures. First, we found that seizures were extremely rare during task engagement, perhaps due to the increased arousal level required to perform the task. Second, treatment with VPA, a common AED, reduced seizures yet had no effect on AET performance. Finally, we observed a lack of any increase in power in seizure-related bands during incorrect trials in *Scn8a*[+/-]mice or on average in comparison to WTs during the AET. Taken together, these findings suggests that the attention impairments and seizures arise from distinct mechanisms.

### Medial PFC PVINs in attentional processing and attention dysfunction

One such mechanism may be the recruitment of mPFC PVINs. The PFC has long been implicated in attention across species. However, only with recent advances has it been possible to begin to answer questions about the contributions of genetically distinct cell-types to attentional performance. Optogenetic (*Kim et al., 2016a*) and pharmacogenetic (*Ferguson and Gao, 2018b*) manipulation of PVINs improves vigilance and attentional flexibility in rodents, raising the question of how PVINs support basic attentional processes. Using a task designed specifically to capture attentional engagement, we observed that by several measures, population rises in PVIN activity were associated with overall accuracy. On a trial-by-trial basis, cue-evoked PVIN activity could be predictive of trial outcome in WTs with the greatest predictive value at the intermediate cue.

Cue-evoked PVIN activity was significantly reduced in *Scn8a*[+/-]mice compared to that of WTs, and this occurred with impairments in AET performance. This was corroborated by network electrophysiology experiments both in vitro and in vivo, providing a novel circuit-level mechanism for attention impairments in absence epilepsy. Reductions in feed-forward putative inhibition and high-frequency PVIN output to pyramidal neurons in vitro were found concomitant with a loss of gamma power during the AET, arguing for PVIN-mediated pathology in *Scn8a*[+/-]mice (*Cardin et al., 2009*; *Sohal et al., 2009*). Broadly alterations in gamma activity were linked to both the attention deficits and restoration in *Scn8a*[+/-]mice. However, a discrepancy was found between the observation of the ability of 40 Hz gamma stimulation to improve attention performance and the loss of high-gamma (60–90 Hz) in *Scn8a*[+/-]mice during the AET.

What constitutes high versus low gamma varies, with some reports classifying high as 100–150 Hz (*Jackson et al., 2011*; *Lee and Jones, 2013*), while others make distinctions between low, mid, and

high gamma (*Buzsáki and Wang, 2012*). There is evidence that low and high likely arise from separate mechanisms, due to differential spatial localization (*Belluscio et al., 2012*; *Oke et al., 2010*) and relationships with distinct behavioral states (*Kay, 2003*; *van der Meer and Redish, 2009*). ECoG recordings have the advantage of reflecting a larger spatial range of neural activity compared to LFP and likely are more reflective of network synchrony (*Ray et al., 2008*). Conversely, ECoG can obscure the ability to observe local rhythms in Layer 2/3 or 5 (*Lee and Jones, 2013*; *Yazdan-Shahmorad et al., 2013*). Slice work from the visual cortex suggests high gamma predominates in superficial layers (*Oke et al., 2010*) raising the question of whether our ECoG findings may indicate a Layer 2/3 specific pathology in high-gamma power. Future studies will utilize simultaneous microelectrode LFP recordings along with surface ECoG during the AET to evaluate how network activity across cortical layers correlates with ECoG, given its higher feasibility for clinical usage in humans.

While our findings of mid/high vs low gamma require future consideration, importantly, hypoactivity of PVINs and AET performance impairments, reductions in gamma power and AET improvement with optogenetic stimulation in *Scn8a*$^{+/-}$mice were all observed at the intermediate cue length. This suggests that response variability at the intermediate cue specifically involves attentional processes linked to PVIN dynamics, gamma activity, and potential synchronicity in this range. Further, the task can be used to continue to elucidate cell-types and circuits supporting attention along with attentional dysfunction.

Questions remain about the source of the recruitment of PVIN activity, contributions of other inhibitory cell types to task components in the AET, and how mPFC output modulates activity in other structures to drive the appropriate behavioral response. Studies point to the involvement of the mediodorsal thalamus (*Delevich et al., 2015*; *Freeman and Nicholson, 1975*; *Rikhye et al., 2018*; *Schmitt et al., 2017*) in regulation of mPFC PVIN activity related to attention, so future studies will examine whether deficits in PVIN activity in absence epilepsy arise locally or result from thalamic hypofunction. Of note, previous work in *Scn8a*$^{+/-}$mice suggests that relay cell function of another thalamic nucleus, the ventrobasal thalamus remains intact (*Makinson et al., 2017*), suggesting that changes in intrinsic excitability of thalamic relay neurons is not strongly affected by partial loss of Scn8a. Regardless of their mechanisms of regulation or recruitment, broadly elevated mPFC PVIN activity seems to provide an optimal cortical state for improved information processing during attention and other higher order cognitive behaviors. This may be due to their ability to suppress neurons representing distracting information (*Ferguson and Gao, 2018a*) and their role in facilitating gamma oscillations which have been linked enhancing neuronal representations and improving information transfer across cortical areas (*Bartos et al., 2007*; *Benchenane et al., 2011*).

## Beyond cognitive deficits in patients with absence epilepsy and animal models

Somewhat surprisingly, our behavioral tests revealed that other behaviors were essentially unaffected. While attention impairments present as one of the most consistent comorbidities, reductions in sociability and affective behaviors have previously been reported in patients (*Rantanen et al., 2012*) and rodent models, including the Wistar Albino Glaxo from Rijswijk (WAG/Rij) and Genetic Absence Epilepsy Rat from Strasbourg (GAERS) rat models (*Henbid et al., 2017*; *Sarkisova and van Luijtelaar, 2011*) and mice with *Scn8a* mutations (*McKinney et al., 2008*; *Papale et al., 2010*). Reports from animal models suggest that ethosuximide, another commonly prescribed AED in absence epilepsy, may mitigate some affective symptoms in models of absence seizures (*Dezsi et al., 2013*) and chronic pain (*Kerckhove et al., 2019*). Given that AEDs can yield improvement across some symptom domains (seizures and affective symptoms), this points toward a likely distinct mechanism of attention and cognitive impairments, as they remained resistant to VPA treatment in *Scn8a*$^{+/-}$mice. Additionally, *Scn8a*$^{+/-}$mice exhibited increases in repetitive responding during the AET which may indicate higher levels of behavioral inflexibility. Behavioral flexibility can present in various ways with distinct circuit contributions (*Hamilton and Brigman, 2015*), thus future studies will examine the impact of *Scn8a* loss on set-shifting and reversal learning along with the potential involvement of the orbital frontal cortex, thalamus, and ventral striatum (*Floresco et al., 2009*). Additional studies are also needed to characterize the generalizability of how blunted PVIN activity contributes to other behavioral impairments in *Scn8a*$^{+/-}$mice and across rodent models.

## Implications for cognitive therapies in absence epilepsy and other disorders

As attentional engagement is a prerequisite for information perception related to other cognitive domains, such as working memory, attentional shifting, and even information encoding for short or long-term memory, deficits in attention are likely to lead to a host of cognitive abnormalities. In addition to absence epilepsy, attention impairments present as a comorbidity in several diseases with complex behavioral phenotypes, including schizophrenia (*Green, 1996*), autism (*Allen and Courchesne, 2001*), depression (*Rock et al., 2014*), bipolar disorder (*Cullen et al., 2016*), Alzheimer's disease (*Perry and Hodges, 1999*), and epilepsy (*Holmes, 2015*). Of note, *Scn8a* mutations are also present in other disorders with cognitive impairments including epileptic encephalopathies, intellectual disability, and autism spectrum disorders (*Butler et al., 2017*; *Larsen et al., 2015*; *Wagnon et al., 2017*). If PVIN hypofunction continues to be linked the etiology of attention and cognitive dysfunction across disease states, moving forward it will be critical to identify strategies for cell-type-specific augmentation of PVIN activity and/or gamma oscillations. This will hopefully accelerate development of better therapies for attention that could improve functional outcome in absence epilepsy and myriad neurological and neurodevelopmental disorders.

# Materials and methods

## Lead Contact

Further information and requests for resources and reagents should be directed to and will be fulfilled by the lead contact, Brielle Ferguson (Brielle.Ferguson@childrens.harvard.edu).

## Materials availability

This study did not generate any unique reagents.

### Key resources table

| Reagent type (species) or resource | Designation | Source or reference | Identifiers | Additional information |
|---|---|---|---|---|
| Strain, strain background (musculus males and females) | B6;129P2-Pvalbtm1(cre)Arbr/J | The Jackson Laboratory | Stock No: 008069; RRID: IMSR_JAX:008069 | |
| Strain, strain background (musculus males and females) | C3Fe.Cg-Scn8amed/J | The Jackson Laboratory | Stock No: 003798; RRID: IMSR_JAX:003798 | |
| Strain, strain background (musculus males and females) | B6.Cg-Tg(Slc32a1-COP4*H134R/EYFP)8Gfng/J (VGAT-ChR2) | The Jackson Laboratory | Stock No:014548 RRID:IMSR_JAX:014548 | |
| Strain, strain background (musculus males and females) | Ai32(RCL-ChR2(H134R)/EYFP) (Ai32) | The Jackson Laboratory | Stock No: 014548 RRID:IMSR_JAX:012569 | |
| Transfected construct (musculus) | AAV-EF1a-DIO-EYFP | University of North Carolina Vector Core | N/A | |
| Transfected construct (musculus) | AAV5-CAG-FLEX-GCaMP6s | Addgene | Cat# 100842-AAV5 | |
| Transfected construct (musculus) | AAV5-FLOX-chR2-mCherry | Addgene | Cat# 20297-AAV5 | |
| Chemical compound, drug | DNQX | Sigma | D0540 | |
| Chemical compound, drug | CPP | Sigma | C104 | |
| Chemical compound, drug | Valproic acid sodium salt | Sigma | P4543 | |
| Software, algorithm | Prism | GraphPad Software Inc | https://www.graphpad.com/scientific-software/prism/ | |
| Software, algorithm | MATLAB | MathWorks | https://www.mathworks.com/ | |

*Continued on next page*

*Continued*

| Reagent type (species) or resource | Designation | Source or reference | Identifiers | Additional information |
|---|---|---|---|---|
| Software, algorithm | GPower | Heinrich Heine Universität Düsseldorf | http://www.gpower.hhu.de/ | |
| Software, algorithm | Spyder | Open Source | https://www.spyder-ide.org/ | |
| Other | DAPI stain | Invitrogen | D1306 | (1 µg/mL) |

## Experimental model and subject details

Mice with the heterozygous loss of function mutation in *Scn8a* were purchased from the Jackson laboratory (C3HeB/FeJ-Scn8amed/J, *Kohrman et al., 1996*, Stock#: 003798), referred to in this manuscript as *Scn8a$^{+/-}$*. For the behavioral experiments, including the attentional engagement task (AET), social interaction, novel object recognition, and the anxiety task, *Scn8a$^{+/-}$* mice were maintained on a C3HeB/FeJ background resulting in litters that were either *Scn8a$^{+/-}$* or WT (*Scn8a$^{-/-}$*). For fiber photometry and optogenetics experiments, *Scn8a$^{+/-}$* mice were crossed with mice homozygous for Cre expression in parvalbumin interneurons (B6;129P2-Pvalbtm1(cre)Arbr/J, PV-Cre, Jax Stock#: 008069) to generate *Scn8a$^{+/-}$* and *Scn8a$^{-/-}$* mice, which were both heterozygous for PV-Cre. For photoinhibition experiments, hemizygous VGAT-ChR2 mice were purchased from Jackson Laboratory (VGAT-mhChR2-YFP, Jax stock#: 014548) and bred with C57BL/6 J mice to maintain the colony (Stock#: 000664). For whole-cell slice recordings and in in vivo optogenetics experiments, Ai32 mice (RCL-ChR2(H134R)/EYFP, Jax stock: 012569) were crossed with PV-Cre;*Scn8a$^{+/-}$* mice to generate *Scn8a$^{+/-}$* and *Scn8a$^{-/-}$* mice expressing ChR2 and parvalbumin interneurons (PV:Ai32-eYFP$^{+/+}$; *Scn8a$^{+/-}$* or PV:Ai32-eYFP$^{+/+}$;WT). PV-Cre;*Scn8a$^{+/-}$*-eYFP$^{-/-}$ mice were also utilized in an optogenetic validation control experiment described below. All pups were genotyped at postnatal day 21 using a commercial service (Transnetyx), and only mice of the desired genotypes were selected and randomly assigned to experimental groups. All mice were maintained on a reverse 12 hr dark/light cycle, and all experiments occurred during their active cycle. Males and females were used all experiments and were 2–4 months of age. No experimental differences due to sex were observed. Food and water were available ad libitum, except for during training and testing for the AET. For the AET and variable delay-to-signal task (VDST), mice were food restricted to approximately 85% of their free-feeding weight, 2 weeks prior to beginning experiments. All experiments were approved by the Stanford Administrative Panel on Laboratory Animal Care (APLAC, Protocol #12363) and were performed in accordance with the National Institute of Health guidelines.

## Method details

### Surgery

Mice aged postnatal day 40–50 were used for all surgeries. For the fiber photometry experiments, *Scn8a$^{+/-}$* or WT mice received a unilateral injection of a AAV5-CAG-FLEX-GCaMP6s (Addgene, #100842-AAV5) tagged with GFP into the prelimbic medial prefrontal cortex (mPFC) using the coordinates (AP: –2.6, ML:+0.25, DV: –1.0). Briefly, a Hamilton syringe was slowly lowered to the target region, and the virus (300 nL) was infused at a rate of 50 nL/min. The cannula remained in place for 5 min post injection, then an optical fiber (400 µm, 0.48-NA, 1.25 mm-diameter stainless steel ferrule, Doric Lenses) was implanted slowly directly above the viral infusion site. For optogenetics, the virus used was an AAV5-FLOX-ChR2-mCherry (Addgene, #20297-AAV5) or a AAV5-Ef1a-DIO-eYFP (UNC Vector Core). The viral injection and optical fiber (200 µm, 0.22-NA, 1.25 mm-diameter stainless steel ferrule, Doric Lenses) insertion procedure was the same, except it was bilateral rather than unilateral. VGAT:ChR2 mice also were implanted with bilateral optical fibers into the mPFC. Following optical fiber insertion, the optical fiber was stabilized, and the incision was closed using dental cement. After surgery, animals recovered for one week before food restriction, and a week later, experiments began. Following experiment completion, animals were sacrificed for histological verification of viral expression and optical fiber placement. Animals with insufficient viral expression or improper optical fiber placement were excluded from analysis.

## Behavior

Attentional Engagement Task (AET). For the AET, mice were food restricted for at least one week prior to experiments to increase motivation for the task. First, mice were habituated to the chamber, that was modeled from *Wimmer et al., 2015* and consisted of a center initiation port, and two reward ports on the left and right side. During training, mice learned to use visual cues (a white LED positioned below each reward port) to locate a food reward. During each session, mice remained in the chamber until 80 min had passed, or they had completed 100 trials, whichever came first. To initiate trials, mice learned to place their nose in a center portal, breaking an infrared light-emitting diode (IR LED) beam between an IR LED-photodiode pair. Following the visual cue (5 s LED flash), doors blocking access the two reward ports would raise. A correct choice would trigger reward delivery (10 µl of condensed milk) from a syringe pump (World Precision Instruments) connected to the reward port. Following an incorrect choice, the doors would close, leading to a 30 s timeout, where the animal could not initiate another trial. All mice had to reach a criterion of at least 70% correct for 3 consecutive training days, or 90% correct for 2 days. During testing, after trial initiation, a visual cue of either 5 s, 2 s, 1 s,.5 s, and.1 s, or 5 s, 2 s, or.5 s, indicated the correct location of the food reward. The cue length and correct side varied pseudorandomly, and performance was averaged over 3 test days. Omissions were quantified as trials in which the animal took longer than 5 s following cue termination to make a choice. Reaction Time was the time between trial initiation and making a response in one of the reward ports. In single-cue length blocks, trials were organized on to 10-trial blocks of each cue length from longest to shortest that repeated until the end of the session. Repetitive responding was quantified by counting the number of blocks of five consecutive trials in which the animal selected the same port normalized by the total trial number in a session. For the VGAT:ChR2 experiments, training for the VDST began immediately following the final AET test day. Animals were trained using the same strategy of reinforcement and punishment as described above to respond to fixed 1 s cue following a 2 s delay. To avoid overtraining, mice had to complete only one day at 60% correct to continue to the testing stage. During testing, the delay prior to the cue varied between 3 s, 4 s, and 5 s while the cue length remained fixed at 1. Trial logic was controlled by a microcontroller (Arduino Mega 2650), and custom MATLAB scripts were used to analyze data and determine the percentage correct at each cue length.

Sociability. To measure sociability, mice were tested in a three-chamber social interaction task (*Moy et al., 2004*). For the three-chamber paradigm, the apparatus was partitioned into three zones, with a small walkway so the mouse could move freely between chambers. A camera was mounted directly above the apparatus to capture video that was used for later analysis. Animals were habituated to the entire apparatus for ten minutes. Then, an empty cage was positioned in one chamber, a cage containing a novel mouse was placed in the opposite chamber, and the mouse was given ten minutes for exploration. Videos were scored by the experimenter, and active interaction, time spent oriented toward, sniffing, or interacting with the empty cage versus the cage with the novel mouse was measured.

Object Recognition. Object recognition was tested as previously described (*Barker et al., 2007*) in a clear plastic chamber (14x16 x 6") in which mice were habituated to for 5 min on the day prior to beginning testing. One object pair was a set of four stacked bottle caps, alternating between purple and orange caps. The other was a glass jar of equal height as the tower of caps, filled with purple sand. The testing day consisted of a sample phase and a test phase, with a 2-hr delay between. During the sample phase, mice were given 3 min to explore two identical objects placed in the center of the arena. For the test phase, one of the objects was replaced, and the animal was allowed to explore both objects freely for 5 min, while video was acquired from above the testing arena. The experimenter scored the videos by measuring the total time spent exploring each object within the first 20 s of total exploration. A difference score was calculated by taking the time exploring the object in phase 2 from phase 1, then dividing that difference by the total time (20 s). Heatmaps of the animal's activity in the during object recognition and social interaction using ezTrack in Jupyter Notebooks (*Pennington et al., 2021*).

## Electrocorticography recordings and analysis

Mice were surgically implanted with bilateral ECoG screw electrodes into the skull above the somatosensory cortex (S1) or gold pin electrodes over the mPFC, and recorded channels were referenced

to an ECoG screw electrode above the cerebellum. Data were acquired using the OpenEphys data acquisition board and software (*Siegle et al., 2017*). Data were bandpass filtered using a Butterworth filter (0.1–100 Hz) and sampled at 30 kHz. We recorded ECoG from animals over three sessions where they were initiating trials at a rate of at least 1 trial per minute (engaged in task). For the S1 recordings, this was compared to 3 one-hour home cage sessions. All data processing and analysis were performed using custom MATLAB scripts. Continuous wavelet transformation was used for spectral analysis (*Torrence and Compo, 1998*) as described in *Sorokin et al., 2017*. For seizure detection, we utilized the full power spectrogram to detect seizures and events were only included if they were between 2 and 30 s in duration. Additionally, events that exhibited maximum power beyond the typical absence seizure spectrums (7–10 Hz) were discarded. Seizures were normalized to session length and displayed as seizures/minute. For power analysis, data was segregated by cue length and accuracy, and mean power in the modified theta (Ø, 7–10 Hz) or β (10–20 Hz) range which encompass the fundamental and second harmonic frequency bands for absence seizures (*Sorokin et al., 2017*) was extracted, and displayed as mean power and variance in relation to the cue.

For the mPFC recordings, we extracted data from the cue period across trials and separated data by cue length and accuracy. We generated ECoG power spectrograms (1–100 Hz) using the continuous wavelet transform (Morlet wavelet in the MATLAB wavelet toolbox). The spectrograms were normalized to peak and $\log_{10}$ transformed, then power was averaged within bands of interest (4-8, 8-12, 12-30, 30-50, 60-90).

## Slice electrophysiology

Extracellular Multi-Unit. To measure differences in network activity, extracellular recordings were obtained from mice who had been previously used for fiber photometry experiments. Mice were anesthetized with pentobarbital (50 mg/kg) and perfused with ice-cold sucrose buffer containing (in mM): 234 sucrose, 2.5KCl, 1.25 $NaH_2PO_4$, 10 $MgSO_4$, 0.5 $CaCl_2$, 26 $NaHCO_3$, and 11 glucose, equilibrated with 95% O2 and 5% CO2, pH 7.4. Mice were decapitated, the brain was extracted, and coronal slices (400 µm) were collected using a Leica VT1200S vibratome. Slices were transferred to oxygenated ACSF (containing in mM: 126 NaCl, 2.5 KCl, 1.25 $NaH_2PO_4$, 2 $MgCl_2$, 2 $CaCl_2$, 26 $NaHCO_3$), that bubbled continuously with 95% O2 and 5% $CO_2$, and incubated at 32 °C for 1 hr. Then, slices were incubated at room temperature for 1–5 hr before they were placed in an interface recording chamber, in which they were continuously perfused with oxygenated ACSF (30–32°C) at a flow rate of 2–3 ml/min. A linear silicon multichannel probe (16 channels, 100 µm inter-electrode spacing, NeuroNexus Technologies) was placed in the prelimbic mPFC perpendicular to the laminar plane, such that the electrode array collected local field potentials (LFPs) from each cortical lamina. A bipolar tungsten electrode was placed immediately ventral to the recording array in layer II/III to deliver electrical pulses (0.1ms, 100–500 µA in 100 µA increments) to evoke synaptic responses. Signals from all sixteen channels were digitized at 25 kHz, using a 3000 Hz lowpass filter, amplified and stored using a RZ5D processor multichannel workstation (Tucker-Davis Technologies). To better examine the location, direction, and magnitude of currents evoked in response to electrical stimulation, a current source density (CSD) analysis was performed by calculating the second spatial derivative of the LFP (*Freeman and Nicholson, 1975*). When net positive current enters a cell, this creates an extracellular negativity that is reflected in a current 'sink', and appears as a negative deflection in the CSD. Conversely, current 'sources' indicate net negative current flowing into a cell and will create positive CSD responses. DNQX (25 µM, Sigma, D0540) and CPP (1 µM, Sigma, C104) were used to determine disynaptic and post-synaptic components. Peak amplitudes were quantified using trial-averaged responses at each stimulation intensity (10 trials each). LFP and CSD plots were generated using custom MATLAB scripts.

Whole-Cell Patch Clamp. Slices were collected as described above for LFP recordings, with slice thickness of 300 µm from adult animals (P60-90). Pipette solutions contained (in mM): 120 K-gluconate, 11 KCl, 1 MgCl2, 1 CaCl2, 10 HEPES, 1 EGTA, and pH was adjusted to 7.4 with KOH (290 mOsm). For VGAT:ChR2 continuous light experiments, we generated a noisy current injection of 5 s, 2 s, and 0.5 s lengths to mimic naturalistic synaptic input using a stimulus waveform generator (noise waveform, 150 pA DC, 10 pA white gaussian noise stim amplitude, 3ms alpha tau, 100 Hz noise filter, *Mainen and Sejnowski, 1995*). This allowed us to evoke spikes that were variable throughout the traces but consistent across trials. We recorded evoked spikes from Layer 2/3 pyramidal neurons in current clamp mode and alternated between no light and continuous light (GABAergic inhibition) conditions during

the current injection (5 trials of each per cell). We also recorded the membrane potential (mV) change in pyramidal neurons in response to continuous blue light (5 trials). To measure whether continuous light could provide sustained inhibition, we restricted our analysis window to the last.5 s bin for our measurements of spike counts and membrane potential. Spikes were detected using a thresholding procedure, and membrane potential was averaged across the.5 s bin and compared to the 0.5 s prior to light stimulation (baseline). For the PVIN optogenetic whole-cell experiments, mPFC slices were collected from PV:Ai32-*eYFP$^{+/+}$;Scn8a$^{+/-}$*or PV:Ai32-*eYFP$^{+/+}$;*WT mice with endogenous expression of ChR2 in PVINs. Evoked inhibitory post-synaptic potentials (IPSP) trains were recorded from Layer 2/3 pyramidal neurons in current clamp while providing a 0.5 s duration 40 Hz TTL train to trigger blue LED light pulses (5ms pulse length). To determine failure rate, we aligned each response to the TTL pulse recorded in Clampfit (indicating the signal to the LED driver) and measured the peak response compared to the baseline within a 0.025 s window. Responses smaller than 0.25 mV were considered failures. Then, we found the percentage of failures within each train and averaged that across 5 trials. To look at PVIN regulation of pyramidal neuron spiking we delivered a 0.5 s 40 Hz train and simultaneous noisy current injection with parameters described above and measured spiking in Layer 2/3 pyramidal neurons with and without the light stimulus train. Spikes were again detected using a thresholding procedure and compared within cells between light and no light conditions to generate a spike reduction rate (number spikes no light - number spikes light / number spikes in light). No difference in resting membrane voltage, membrane capacitance, or series resistance was observed between any control and experimental groups.

## Fiber photometry

Fiber Photometry was used to measure aggregate Ca$^{2+}$ signals from GCaMP6s-expressing PVINs during the AET. The set up was designed as described in *Kim et al., 2016a*. Briefly, an implanted optical fiber (1.25 diameter stainless steel ferrule) was coupled to a patchcord with ceramic sleeves (Thorlabs, ADAL1). The patchcord terminated in an SMA connector (Thorlabs, SM1SMA), which was focused onto the sensor of an sCMOS camera (Hamamatsu, ORCA-Flash4.0). A series of emission filters and dichroic mirrors were arranged such that 470 or 405 nm light from two LEDs could be transmitted through the patchcord, and 535 nm light would be collected at the working distance of the objective. A custom MATLAB GUI (*Kim et al., 2016c*; *Kim et al., 2016b*) triggered the LED drivers (Thorlabs, LEDD1B) through a National Instruments Data Acquisition device (NIDA, PCIe-6343-X). This GUI triggered alternating pulses of the GCaMP activating or control wavelength (470 and 405 respectively) at a frequency of 40 Hz to result in collection of Ca2 +and isosbestic signals at a 20 Hz sampling rate. To account for motion-artifacts, the reference trace (405 nm) was scaled to the GCaMP (470 nm) using a least-squares regression, then we subtracted the scaled reference from the GCaMP signal to get a normalized trace. To obtain the change in fluorescence over time (dF/F), we subtracted the median of the normalized 470 nm signal from the continuous signal at each point in time, divided by the median (f(continuous) - f(median)/f(median)). To align the dF/F to behavior, the Arduino sent TTL pulses at the start of each trial to the NIDA device for storage in a MATLAB file along with the fiber photometry data. The dF/F was separated into trials by cue length and accuracy, and peak, average, and time-to-peak were quantified using custom MATLAB scripts. Representative heatmaps of individual trial data were generated using the MATLAB function imagesc.

## Modeling

Trial Prediction Analysis was performed in python 3.8 using a combination of the *numpy, scipy, sklearn, pandas, imblearn,* and *xgboost* toolkits. Groups were separated, blinded, and trials were randomly selected from each group for model training. Extreme outliers which were classified as a signal with a mean standard deviation greater than ten times that of the group average. Feature generation was run over all timeseries using simple summary statistics from the *numpy* and *pandas* toolkits. All features (excluding maximum and minimum values of timeseries) were min/max normalized. Given that there were often much more correct than incorrect trials at longer cue lengths, the *imblearn* implementation of Synthetic Minority Oversampling Technique (*Chawla et al., 2002*) and Tomek-Links-based under sampling (*Tomek, 1976*) was used to achieve a near 50:50 class balance across each group. The data was modeled via an optimized implementation of Extreme Gradient-Boosted Classification and Regression Trees (XGBoost from the *xgboost* toolkit). Trials and features were randomly sampled

(fraction = 80% and 30% respectively) every iteration of XGBoost in order to combat over-fitting. Model performance was evaluated via the average receiver operating characteristic-area under the curve (ROC-AUC) score across a repeated (k=10, n=3), stratified, k-fold cross-validation mechanism from the *sklearn* toolkit. ROC-AUC curves were further compared to a null-model (XGBoost trained on a shuffled training dataset).

## Optogenetics

To deliver optogenetic stimulation during behavior, we utilized several components of the fiber photometry system described above including the LED and LED driver, patchcord, and optical fibers. Once the animal initiated a trial, the Arduino sent TTL pulses directly to the LED driver, such that blue light pulses (470 nm, 40 Hz, 5ms pulse width, 5 mW) were delivered to through the patchcord to the mPFC throughout the duration of the cue.

## Optogenetics with electrocorticography

PV:Ai32-eYFP$^{+/+}$;Scn8a$^{+/-}$ mice or PV:Ai32-eYFP$^{-/-}$;Scn8a$^{+/-}$ were implanted unilaterally with a gold pin for recording ECoG along with a reference screw in the cerebellum. An optical fiber was implanted in the opposite cortex at a 15 degree angle for delivering 40 Hz trains at lengths of 5 s (long), 2 s (Int), and.5 s (Short). Five trials of each train length were recorded with a 30-second intertrial interval in each animal's home cage. Power in each frequency band was extracted as described above and compared to baseline (0.5 s) prior to the stimulus train. Rel. power [log$_{10}$] was calculated as described above and compared between baseline and light conditions.

## Pharmacology

For evaluation of VPA treatment on AET performance and seizures, animals were treated twice daily with the following schedule: Twice daily saline beginning in the evening 2 days before testing, and continuing for two days of testing, then twice a day VPA (200 mg/kg) beginning two evenings before day one of VPA testing and continuing for three days. ECoG was recorded during each behavioral session. Data was analyzed from 2 days of saline testing and 3 days of VPA testing and averaged for comparison. Animals were removed from food restriction and allowed a 2-week wash out period, then this same approach was repeated without behavioral testing. Animals were placed in the behavioral chamber for 1-hr ECoG sessions while satiated. Seizures were detected as described above in each session, averaged over sessions (two saline and three VPA), and displayed as seizures/minute in saline versus VPA.

## Histology

All animals with exception of those for electrophysiological recordings were sacrificed for histological confirmation of viral injection and expression. Mice were transcardially perfused with 0.1 M PBS followed by 4% paraformaldehyde (PFA) in PBS. Brains were removed and postfixed for 48 hr, then cryoprotected in 10% to 30% sucrose solution for an additional 48 hr. Forty μm sections were collected using a vibratome, and sections were mounted and cover slipped with VECTASHIELD Antifade Mounting Medium with DAPI (Vector Laboratories, Burlingame, CA). Images were acquired with a ZEISS ApoTome 2.

## Quantification and statistical analysis

All statistical analysis was performed using MATLAB, Prism 6 (GraphPad), and python using the Spyder IDE. We determined sample sizes using a power analysis along with our previous studies and literature. All measures are presented as mean ±SE (mean represented with a line and average point in bold with individual data points and SE box at 50% transparency) or individual data points. For all behavior experiments, n represents individual animals. In slice experiments, n represents number of slices with no more than 2 slices per animal. For fiber photometry, electrocorticography (ECoG), and optogenetics experiments, n represents AET data averaged over 3 days from individual animals. Normality was determined using the Shapiro-Wilk test to direct the usage of parametric versus nonparametric tests. Outliers were determined using the mean absolute deviation method and any data set where outliers were identified is noted in the source data files. For normal data, the comparisons of two independent groups were done using a Student's t test, while paired samples were compared

using paired t tests. For data without a normal distribution, a Mann-Whitney U test (independent) or Wilcoxon Signed Rank test (paired) was used. For comparisons of multiple groups and cue lengths, a two-way ANOVA was used followed by post-hoc Holm-Sidak's test with correction for multiple comparisons. $p < 0.05$ was considered significant and was represented by a one asterisk ('*'), while $p < 0.01$ and 0.001 was represented with two and three asterisks, respectively ('**' and '***'). The experimenter was blinded to group identity during data all data collection and analysis.

## Source data files

Source data files for *Figure 1*, *Figure 2*, *Figure 3*, *Figure 4*, *Figure 5*, *Figure 6*, *Figure 7*, and *Figure 8*, along with , *Figure 3—figure supplements 1 and 2*, *Figure 4—figure supplement 1*, *Figure 6—figure supplements 1 and 2*, , and *Figure 7—figure supplements 1 and 2* are available at the following link: https://doi.org/10.7910/DVN/HYY2MB.

## Acknowledgements

BRF is supported by the Stanford School of Medicine Dean's Fellowship, NIH-NINDS T32NS007280-35, NIH-NINDS Diversity Supplement to R01NS034774-24A1, and NIH-NINDS F32NS112764. JRH is supported by the NIH-NINDS R01NS034774-24A1 and NIH-NINDS T32NS007280-35. We thank Dr. Christina Kim for her consultation on the set up and implementation of the fiber photometry microscope and data analysis. We thank Dr. Ralf Wimmer for his consultation on the design and implementation of the behavior chamber and training paradigm for the attentional engagement task. We thank Dr. Wen-Jun Gao for his critical feedback on the manuscript. We thank Revathi Kaduru for her assistance in the electrocorticography experiments.

## Additional information

### Competing interests

John R Huguenard: Reviewing editor, *eLife*. The other authors declare that no competing interests exist.

### Funding

| Funder | Grant reference number | Author |
|---|---|---|
| National Institute of Mental Health | 1K99MH128892 | Brielle Ferguson |
| National Institute of Neurological Disorders and Stroke | 1F32NS112764 | Brielle Ferguson |
| Stanford University | Dean's Postdoctoral Fellowship | Brielle Ferguson |
| National Institute of Neurological Disorders & Stroke | 5R01NS034774 | John R Huguenard |

The funders had no role in study design, data collection and interpretation, or the decision to submit the work for publication.

### Author contributions

Brielle Ferguson, Conceptualization, Data curation, Formal analysis, Funding acquisition, Validation, Investigation, Visualization, Methodology, Writing – original draft, Project administration, Writing – review and editing; Cameron Glick, Provided revised modeling analysis and figures during the second revision; John R Huguenard, Conceptualization, Supervision, Funding acquisition, Visualization, Project administration, Writing – review and editing

### Author ORCIDs

Brielle Ferguson http://orcid.org/0000-0001-9210-6504

John R Huguenard [ID] http://orcid.org/0000-0002-6950-1191

### Ethics

All experiments were approved by the Stanford Administrative Panel on Laboratory Animal Care (APLAC, Protocol #12363) and were performed in accordance with the National Institute of Health guidelines.

### Decision letter and Author response

Decision letter https://doi.org/10.7554/eLife.78349.sa1
Author response https://doi.org/10.7554/eLife.78349.sa2

## Additional files

### Supplementary files

• MDAR checklist

### Data availability

Source data to recreate figures presented in this manuscript has been uploaded to Dataverse and can be found here: https://doi.org/10.7910/DVN/HYY2MB. Custom MATLAB scripts used for analysis are available at https://github.com/huguenardlab/fiberphotometry, (copy archived at *Huguenardlab, 2023*).

The following dataset was generated:

| Author(s) | Year | Dataset title | Dataset URL | Database and Identifier |
| --- | --- | --- | --- | --- |
| Ferguson B | 2023 | Prefrontal PV interneurons facilitate attention and are linked to attentional dysfunction in a mouse model of absence epilepsy | https://doi.org/10.7910/DVN/HYY2MB | Harvard Dataverse, 10.7910/DVN/HYY2MB |

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
