## [Editor Report]

This study reports on the circuits contributing to impairment in attention in absence epilepsy linked to reduced Scn8A expression. Using a novel attentional engagement task, the evidence supporting the main conclusions is solid and well-sampled. The results provide a starting point for future experiments to assess functional impairments in parvalbumin-positive interneurons and gamma activity with attention in models of absence seizures and mental health disorders.

---

## [Decision Letter]

**Decision letter after peer review:**

Thank you for submitting your article "Prefrontal PV interneurons facilitate attention and are linked to attentional dysfunction in a mouse model of absence epilepsy" for consideration by *eLife*. Your article has been reviewed by 2 peer reviewers, and the evaluation has been overseen by a Reviewing Editor and Gary Westbrook as the Senior Editor. The following individual involved in review of your submission has agreed to reveal their identity: Qian-Quan Sun (Reviewer #1).

Essential revisions:

*Reviewer #1 (Recommendations for the authors):*

(1) The abstract could be improved: The novel behavior paradigm the authors developed based on previous cue-based tasks is crucial for the author's claim however, they did not describe this in the abstract. They also did not mention loss of gamma power and the gamma-stimulation of PV neurons which seems to be important as well. The methods and background statement could be reduced or eliminated if they choose to. The word " recruitment" is vague, as it did not provide the context of how the PV neurons were recruited.

(2) Results/Figure

(a) A letter was missing for panel A (scale bar mV).

(b) Figure 1. To make the label consistent, 0.5 and 0,1s is better than 500ms and 100ms.

(c) Figure 2 G. The scale is too high (2.5*106) such that the traces can be hardly seen.

(d) The subtitle "AET performance deficits are unrelated to seizure activity in scn8a+/- mice" is somewhat misleading: the title might be interpreted as history of seizures in different scn8a mice, however, the authors measured acute EEG and performed seizure detection and correlate seizure with correct vs. incorrect trials. Consider modifying the subtitle to reflect what they are trying to say.

(3) Discussions:

Since the authors place their studies on the aspects of attention deficits related to absence seizures, some discussion clinical relevance of loss of function of scn8a mutation with the absence epilepsy is expected.

*Reviewer #2 (Recommendations for the authors):*

1. In order to address major point 1 with the existing data, try maximizing the statistical power by using a mixed model approach with trials as the repeated measure (mixed model needed because the number of trials differs per animal). You can find more information here: https://www.theanalysisfactor.com/six-differences-between-repeated-measures-anova-and-linear-mixed-models/.

2. Demonstrate that 40 Hz stimulation increases high gamma band power in the gold pin.

3. Analyze the seizure activity in the mpfc gold pin.

4. Some neural data to demonstrate that 0.5 mW, continuous stimulation is effective would strengthen the paper.

5. The in vitro electrophysiology data could probably all go in to supplement.

6. Typo: "Gender", should be "sex".

7. The clonazepam and supplemental slice physiology experiments do not have N's or statistics.

8. The authors should discuss the findings in the context of Cho et al. 2020 https://www.nature.com/articles/s41593-020-0647-1 who show that trial by trial PV activity impacting cognitive performance.

9. Quantify the currently vague criteria for discarding high power events "beyond the typical".

[Editors' note: further revisions were suggested prior to acceptance, as described below.]

Thank you for resubmitting your work entitled "Prefrontal PV interneurons facilitate attention and are linked to attentional dysfunction in a mouse model of absence epilepsy" for further consideration by *eLife*. Your revised article has been evaluated by Gary Westbrook (Senior Editor) and a Reviewing Editor.

The manuscript has been improved but there are some straightforward issues raised by one of the reviewers that we would like you to address, as outlined below:

*Reviewer #1 (Recommendations for the authors):*

The revision completely and thoroughly addressed all of my previous concerns. The revised manuscript, as it stands, is acceptable for publication at *eLife*.

*Reviewer #2 (Recommendations for the authors):*

The authors have addressed some, but not all, of my concerns,

1. As I wrote in my initial review, collapsing animals across genotypes for regression of PV activity vs behavior when there are behavioral differences across genotypes is statistically invalid. See https://pubmed.ncbi.nlm.nih.gov/27843795/ This same concern applies to machine learning. Addressing this statistical concern is crucial for making one of the key claims of the paper, that changes in PV activity correlate with behavior. As it stands, all that can be claimed is that there are group differences in both behavior and PV activity between the wt and scn8a+/- mice. This issue affects figure 6, E, F, G, and I and corresponding supplements 1-3

2. I understand that the authors want to reserve characterizing the clonazepam for another manuscript. However, they cannot make claims based on single example data. If they do not wish to provide group data and corresponding statistics, they can remove the data and claims about clonazepam from the manuscript.

Typo: line 189 clean copy "less than 1%" should be "fewer than 1%".

Typo: line 254 clean copy "thoe" should be "the".

Please also address the additional points below: 1. Figure 1B add a p-value for 5s.

2. Figure 1 E, remove the extra semicolon at the end of the sentence.

3. In line 164 (attentional load rather than an effect of the loss of Scn8a on visual perception or discrimination), please replace "loss of" with reduced.

4. Please double-check the statistics associated with Figure 3D. Is the p-value listed for the short interval from a post hoc test or does it refer to an effect of group on performance? If the latter, please remove the * and clearly indicate the group effect in the legend. If it is a post hoc test, please also state the test used.

5. Line 250 should be amended to (observed a small reduction in performance with continuous light stimulation (Figure 3D) without a significant).

6. Depending on the result of your statistical assessment of data for Figure 3D please amend the figure title (Photoinhibition of mPFC during cue presentation reduces accuracy in the AET but not the VDST) accordingly.

7. The use of "critical" in line 271 (time. This suggests that the AET is linked to the mPFC, and that mPFC activation is critical during cue) is overstated, especially since performance appears to remain above chance level. Please replace it with "plays some role" or "functions" or another appropriate term.

8. Same comment for line 275 (time. This suggests that the AET is linked to the mPFC, and that mPFC activation is critical during cue). The use of "critical" in line 275 is overstated, as % accuracy is minimally affected and remains above chance performance.

9. On line 319 (endogenously in PVINs (PV:Ai32; Scn8a+/- or PV:Ai32;WT). Given the link between PVINs and gamma), please correct the typo.

10. In figure 4I, please remove references to the clonazepam data as this is an n=1 or increase the number of animals to at least 3.

11. Data in figures 6E-G, the data needs to be separated by genotype and analyzed as such. If the previous data conclusions cannot be consistently reached for data analyzed this way, then please remove claims about the PV activity predicting behavioral response.

12. In Figure 6E, please separate the data by genotypes and analyze it accordingly. Incorrect trials could be included in a separate figure. It is very difficult to see the data points of each genotype in the current form. Is the P-value (p < 0.0001) from a post hoc test? Please list the test.

13. In Figures 6F and G, only 9 points are represented. One point appears to be missing (see Figure 6E where n=10).

14. In Figure 6H, there seems to be more than 5 points from each genotype. What is the source of the additional data? Please clarify in the text.

15. If the training data was randomly selected, please consider revising lines 396-397 to state:

We trained a classifier using randomly selected, training data with the features peak amplitude and time to peak from all trials at each cue length, and evaluated its performance on a subset of held-out data.

---

## [Author Response]

Essential revisions:Reviewer #1 (Recommendations for the authors):(1) The abstract could be improved: The novel behavior paradigm the authors developed based on previous cue-based tasks is crucial for the author's claim however, they did not describe this in the abstract. They also did not mention loss of gamma power and the gamma-stimulation of PV neurons which seems to be important as well. The methods and background statement could be reduced or eliminated if they choose to. The word " recruitment" is vague, as it did not provide the context of how the PV neurons were recruited.

We now mention the task is novel and expand on the task design as the reviewer suggested. We also include the reduction in gamma power along with the rescue using the gamma frequency stimulation in the description of the findings. An excerpt is below:

“Attention function was measured using a novel visual attention task where a light cue that varied in duration predicted the location of a food reward. In *Scn8a^+/-^* mice, we find decreases in parvalbumin interneuron (PVIN) activity in the medial PFC (mPFC) in vitro and PVIN hypoactivity along with reductions in gamma power during cue presentation in vivo*.* We observed that low levels of mPFC PVIN activity correlated with and were predictive of poorer attention performance across animals while gamma-stimulation of PVINs could rescue performance in *Scn8a^+/-^* mice.”

(2) Results/Figure(a) A letter was missing for panel A (scale bar mV).(b) Figure 1. To make the label consistent, 0.5 and 0,1s is better than 500ms and 100ms.(c) Figure 2 G. The scale is too high (2.5*106) such that the traces can be hardly seen.(d) The subtitle "AET performance deficits are unrelated to seizure activity in scn8a+/- mice" is somewhat misleading: the title might be interpreted as history of seizures in different scn8a mice, however, the authors measured acute EEG and performed seizure detection and correlate seizure with correct vs. incorrect trials. Consider modifying the subtitle to reflect what they are trying to say.

(A) We weren’t sure which figure panel this referred to, but we have checked each figure with a mV scale bar and have ensured that each has a panel label and scale bar.

(B) This has been updated in all figures and the text.

(C) We chose the scale in G to match with that in H, where we have shown an example of power detected during an example seizure. The traces in G are barely visible to illustrate that the power detected here is not comparable to that seen during a seizure. We now include an inset scaled up in amplitude to make that clearer.

(D) We have updated the subtitle to the following, “AET performance deficits are unrelated to acute seizure activity in Scn8a^+/-^ mice. “

(3) Discussions:Since the authors place their studies on the aspects of attention deficits related to absence seizures, some discussion clinical relevance of loss of function of scn8a mutation with the absence epilepsy is expected.

A brief discussion on the specific syndromes linked to *Scn8a* is now included. It reads, “Loss of function mutations in *Scn8a* have been observed in patients with absence epilepsy, intellectual disability, attention-deficit hyperactivity disorder, and autism spectrum disorders (Trudeau et al. 2006, Wagnon et al. 2017, Liu et al. 2019). Across these syndromes, attention impairments represent a convergence point, and gaining a better understanding of attentional dysfunction is critical for improving quality of life in patient populations.”

Reviewer #2 (Recommendations for the authors):1. Analyze the seizure activity in the mpfc gold pin.

Current data in our lab (personal observations and a forthcoming manuscript) describe in detail the near simultaneous observation of seizure onset and frequency in all frontoparietal areas including S1 and mPFC. As this is a primary focus of this future manuscript we chose not to focus on or describe this in detail here. We do consistently observe that the seizures appear almost simultaneously in both frontal cortices during quiet waking and similar to S1, little to no seizure activity is observed in mPFC during the task or during trials.

2. Some neural data to demonstrate that 0.5 mW, continuous stimulation is effective would strengthen the paper.

The reviewer is referring to the ability of continuous light, as was used in our VGAT-ChR2 experiments to demonstrate that mPFC disruption could alter performance on the AET. To address this, we conducted a whole-cell physiology experiment where we used continuous light stimulation in brain slices from VGAT-ChR2 mice and measured its effectiveness on reducing spikes induced in pyramidal neurons from a noisy current injection. Light was given continuously for 5s, 2s, and.5s to mimic the cue length conditions in vivo. We observed that spikes were reduced on average for each cue length. We also found that for light stimulation that mimicked that of the long 5s and short 2s cues, spikes were reduced throughout, even at the end of the light stimulus as defined as the last 0.5 seconds. In other experiments, we observed reductions in membrane potential throughout the light stimulation. This data is presented below and now included as two supplemental figures to figure 3.

3. The in vitro electrophysiology data could probably all go in to supplement.

We understand the reviewer’s sentiment, but the goal of this figure was to show in the context of the larger paper that we’ve explored the network from a systems to cell level and back again. We felt that this figure was an important network level step to prompt further investigation into inhibitory neurotransmission and PVINs during behavior. To further motivate these experimental results, we also looked at PVIN output to pyramidal neurons during whole-cell recordings. Together, Figure 4 (network in vitro electrophysiology data) and Figure 5 (whole-cell physiology data) provide an even stronger motivation for examining PVIN activity in vivo during behavior. An excerpt from the revised text is included below along with the new Figure 5 and legend.

“Given PVINs link to driving network activity in the gamma range (Cardin et al., 2009a; Sohal et al., 2009a), we stimulated PVINs with brief.5s 40 Hz blue light trains and recorded the evoked inhibitory post-synaptic potential (IPSP) train in pyramidal neurons at rest in current clamp mode (Figure 5B). We measured synaptic failures by looking at the percentage of IPSPs within the train that were smaller than a.25 mV threshold to determine a failure rate for each neuron. We recorded significantly more failures in Scn8a+/- mice in comparison to WT (Figure 5 C). Additionally, to look at whether gamma-frequency activation of PVINs was capable of regulating pyramidal neuron spiking, we gave a.5s noisy current injection as described above to evoke spikes, and measured the spike number without light in comparison to that recorded with a 40 Hz stimulus train (Figure 5D). We observed that the spike reduction rate with 40 Hz PVIN activation was not changed in Scn8a+/- mice (Figure 5E), suggesting that PVINs are still capable of inhibiting pyramidal neuron output when stimulated optogenetically.”

4. Typo: "Gender", should be "sex".

This has been corrected.

5. The clonazepam and supplemental slice physiology experiments do not have N's or statistics.

The clonazepam experiments have been validated in detail in another forthcoming manuscript from the lab currently in revisions. For the supplemental slice physiology experiment, we have now included a quantification for the DNQX and CPP effects on the L2/3 source in Figure 4-FigSupplement.

6. The authors should discuss the findings in the context of Cho et al. 2020 https://www.nature.com/articles/s41593-020-0647-1 who show that trial by trial PV activity impacting cognitive performance.

We have now included a brief discussion on this topic, “Recent work has shown that in synchronization of gamma activity across hemispheres by PVINs is also critical for attention, as delivering out of phase optogenetic stim can disrupt performance during an attentional flexibility task (Cho et al. 2020). It remains unclear how cross-hemispheric synchrony or the lack thereof may function in our observed deficits but given the propensity for the altered Scn8a network to hypersynchrony, we might speculate that this particular mechanism of network dysfunction might be unaffected in *Scn8a^+/-^* mice.”

7. Quantify the currently vague criteria for discarding high power events "beyond the typical".

This has been clarified. It now reads, “… events that exhibited maximum power beyond the typical absence seizure spectrums (7-10 Hz) were discarded.” Peak power outside of this range does not reflect the absence seizure spectrum and largely represent artifacts due to movement or other non-biological electrical noise.

[Editors' note: further revisions were suggested prior to acceptance, as described below.]

The manuscript has been improved but there are some straightforward issues raised by one of the reviewers that we would like you to address, as outlined below:Reviewer #2 (Recommendations for the authors):The authors have addressed some, but not all, of my concerns,1. As I wrote in my initial review, collapsing animals across genotypes for regression of PV activity vs behavior when there are behavioral differences across genotypes is statistically invalid. See https://pubmed.ncbi.nlm.nih.gov/27843795/ This same concern applies to machine learning. Addressing this statistical concern is crucial for making one of the key claims of the paper, that changes in PV activity correlate with behavior. As it stands, all that can be claimed is that there are group differences in both behavior and PV activity between the wt and scn8a+/- mice. This issue affects figure 6, E, F, G, and I and corresponding supplements 1-3.

We appreciate this insight. We do think our results suggest that PV activity is correlated with behavior, even if we don’t at this point know whether this is causative. We look forward to exploring the correlations between these measures of PVIN activity and performance in follow up studies. Meanwhile, we have taken the larger point that the genotypes should be treated independently. We have removed 6E and 6F and revised Figure 6 supplement 1. We have re-analyzed the data for 6G, and this is discussed in more detail below.

2. I understand that the authors want to reserve characterizing the clonazepam for another manuscript. However, they cannot make claims based on single example data. If they do not wish to provide group data and corresponding statistics, they can remove the data and claims about clonazepam from the manuscript.

The Clonazepam representative has been removed from the manuscript. We have also modified the text to reflect that the LFP results broadly indicate a change in post-synaptic signaling in mPFC, and that a component of this is likely feed-forward inhibition.

Typo: line 189 clean copy "less than 1%" should be "fewer than 1%".

Corrected.

Typo: line 254 clean copy "thoe" should be "the".

Corrected.

Please also address the additional points below:1. Figure 1B add a p-value for 5s.

The bracket and star next to 5s denotes a group difference while the starts under the line graph at (2s and 1s) denote significance determined after post hoc tests. Throughout the paper we included all group p values and then post hoc multiple comparison p values when they were significant. Apologies if this was unclear. The p value for 5s was 0.7399 and the multiple comparison stats are all included below. We have removed the group p value asterisk from the line graph and placed it in the figure legend instead.

**Author response table 1. sa2table1:** 

Sidak’s multiple comparisons test	Mean diff.	95.00% CI of diff.	Significant?	Summary	Adjusted P values
					
WT-Scn8a +/-					
5s	4.778	-5.779 to 15.33	No	ns	0.7399
2s	13.76	3.208 to 24.32	Yes	**	0.0049
1s	10.56	0.007996 to 21.12	Yes	*	0.0497
500ms	1.063	-9.493 to 11.62	No	Ns	0.9996
100ms	5.786	-4.770 to 16.34	No	ns	0.5623

2. Figure 1 E, remove the extra semicolon at the end of the sentence.

Corrected.

3. In line 164 (attentional load rather than an effect of the loss of Scn8a on visual perception or discrimination), please replace "loss of" with reduced.

Corrected.

4. Please double-check the statistics associated with Figure 3D. Is the p-value listed for the short interval from a post hoc test or does it refer to an effect of group on performance? If the latter, please remove the * and clearly indicate the group effect in the legend. If it is a post hoc test, please also state the test used.

The p value refers to the effect of group on performance. We moved the star from the short interval to figure legend to better indicate a group effect and have stated this clearly in the figure legend. We re-ran the stats and included them below:

Two-way RM ANOVA, group effect F_(1,18)_ = 6.990, p = 0.0165, n = 6.

5. Line 250 should be amended to (observed a small reduction in performance with continuous light stimulation (Figure 3D) without a significant).

We added in small, as this is a small change, but it is significant.

6. Depending on the result of your statistical assessment of data for Figure 3D please amend the figure title (Photoinhibition of mPFC during cue presentation reduces accuracy in the AET but not the VDST) accordingly.

This was left as originally written (see response to point 4 above).

7. The use of "critical" in line 271 (time. This suggests that the AET is linked to the mPFC, and that mPFC activation is critical during cue) is overstated, especially since performance appears to remain above chance level. Please replace it with "plays some role" or "functions" or another appropriate term.

This has been corrected and revised to say the following, “This suggests that the AET is linked to the mPFC, and that mPFC activation plays a role during cue presentation in an attention-dependent manner.”

8. Same comment for line 275 (time. This suggests that the AET is linked to the mPFC, and that mPFC activation is critical during cue). The use of "critical" in line 275 is overstated, as % accuracy is minimally affected and remains above chance performance.

This has been corrected and revised to say the following, “Our optogenetic experiments indicated the AET is linked to mPFC function, suggesting a potential frontal locus for performance impairments in Scn8a^+/-^ mice.”

9. On line 319 (endogenously in PVINs (PV:Ai32; Scn8a+/- or PV:Ai32;WT). Given thoe link between PVINs and gamma), please correct the typo.

Corrected.

10. In figure 4I, please remove references to the clonazepam data as this is an n=1 or increase the number of animals to at least 3.

This has been removed.

11. Data in figures 6E-G, the data needs to be separated by genotype and analyzed as such. If the previous data conclusions cannot be consistently reached for data analyzed this way, then please remove claims about the PV activity predicting behavioral response.

This individual groups are not sufficiently powered to look at correlations within separate groups so these graphs have been removed with the exception of the classifier data. Here we repeated the analysis on the two groups independently and reached similar conclusions. In both groups activity during the cue could predict choice and the separate ROC curves are presented in figure 6 showing classifier performance across all three cue lengths. We have also included a supplemental figure illustrating the classifier performance one each cue length separately.

12. In Figure 6E, please separate the data by genotypes and analyze it accordingly. Incorrect trials could be included in a separate figure. It is very difficult to see the data points of each genotype in the current form. Is the P-value (p < 0.0001) from a post hoc test? Please list the test.

The data has been separated by genotypes and re-analyzed. We have also now plotted the correct and incorrect data on separate graphs. There was an effect of accuracy, (two-way RM ANOVA F _(1, 45)_ = 10.372, n = 10) and cue (two-way RM ANOVA F _(1, 45)_ = 41.967, n = 10) on PVIN activity. PVIN activity increased during the int cue in correct trials (p = 0.0138 and 0.0276 for PV-Cre;WT and PV-Cre;Scn8a^+/-^ respectively, Holm Sidak post hoc comparison).

13. In Figure 6H, there seems to be more than 5 points from each genotype. What is the source of the additional data? Please clarify in the text.

The graphing template places an average point in the plot to indicate the average and is not an additional data point. We could have made this more clear, and have clarified that in the methods. The methods have been revised to say the following, “All measures are presented as mean ± SE (mean represented with a line and average point in bold with individual data points and SE box at 50% transparency) or individual data points.”

14. If the training data was randomly selected, please consider revising lines 396-397 to state:We trained a classifier using randomly selected, training data with the features peak amplitude and time to peak from all trials at each cue length, and evaluated its performance on a subset of held-out data.

The sentence has been revised to read as suggested.